# Post-Training Neural Network Pruning using Graph Curvature

**Shuhang Tan**                                                              *tans5@rpi.edu*
*Rensselaer Polytechnic Institute*
**Jayson Sia**                                                               *jsia@usc.edu*
*University of Southern California*
**Paul Bogdan**                                                              *pbogdan@usc.edu*
*University of Southern California*
**Radoslav Ivanov**                                                          *ivanor@rpi.edu*
*Rensselaer Polytechnic Institute*

**Reviewed on OpenReview:** https://openreview.net/forum?id=kwACVY73Ug

## Abstract

This paper provides a fresh view of the neural network (NN) pruning problem, through the lens of graph theory. To achieve effective pruning, we aim to identify the main NN data flows and the corresponding NN connections that are most (and least) important for the performance of the full model. Unlike the standard approach to NN data flow analysis, which is based on information theory, we employ the notion of graph curvature, specifically Ollivier-Ricci curvature (ORC). The ORC has been successfully used to identify important graph edges in various domains such as road traffic analysis, biological and social networks. In particular, edges with negative ORC are considered bottlenecks and as such are critical to the graph's overall connectivity, whereas positive-ORC edges are not essential. We use this intuition for the case of NNs to: 1) construct a graph induced by the NN structure and introduce the notion of neural curvature (NC) based on the ORC; 2) calculate curvatures based on activation patterns for a set of input examples; 3) demonstrate that NC can indeed be used to rank edges according to their importance for the overall NN functionality. We evaluate our method through pruning experiments on a variety of small-to-medium-size models trained on three image datasets, namely MNIST, CIFAR-10 and CIFAR-100. The results indicate that our method can identify a larger number of unimportant edges as compared to existing pruning methods. (Code: https://github.com/SH-Tan/Post-Training-NN-Pruning-using-Graph-Curvature)

## 1 Introduction

Neural networks (NN) have proven to be effective in a variety of areas, such as computer vision (Dosovitskiy et al., 2020), natural language processing (Achiam et al., 2023), and autonomous systems (Hwang et al., 2024). As the popularity of NNs grows, however, so does their size – modern models used in all of the above-mentioned domains often have millions or even billions of parameters Sun et al. (2023). In turn, such models not only contribute to rising energy costs, but they are also more difficult to analyze and repair due to their overwhelming complexity. Thus, there is a growing need for NN pruning approaches that are not only focused on reducing a model's size but also on identifying the main data flows in a pre-trained model.

This paper provides a fresh view of NN pruning by making use of graph theory and network science, which would provide a well-understood tool for compositional analysis of the NN graph structure and corresponding data patterns. In particular, we view a NN as a weighted directed graph, where the neurons are the graph nodes, and the connections are the graph edges. The features that transfer from layer to layer carry the information flow in this graph. Analyzing such an information graph would allow us to identify the main data flow through the NN, which can then be used to prune redundant NN parameters. In future work, such information graphs may be used for a variety of other applications, including NN architecture search or model repair.

The standard approach to NN pruning is through an iterative prune-and-retrain procedure, where unnecessary parameters are pruned, whereas the remaining ones are retrained Han et al. (2015). Classic pruning methods are based on weight magnitude (Han et al., 2015), weight sensitivity (Lee et al., 2018; Santacroce et al., 2023), and loss change (Ma et al., 2023; van der Ouderaa et al., 2023). These techniques' main aim is to prune a NN architecture to a smaller one with similar performance to that of the original. There also exists more recent work on pruning pre-trained large language models (LLMs), by exploiting the parameters' Hessian matrix properties Frantar & Alistarh (2023) or pruning based on weights and activations Sun et al. (2023). In contrast, our paper aims to achieve more effective pruning through analyzing the NN data flows using concepts from graph theory and differential geometry.

In the related literature, one approach to analyze the data flow in a NN is through information theory (Khadivi et al., 2016; Shwartz-Ziv & Tishby, 2017; Nasiri, 2020), such as the seminal work on the information bottleneck principle (Tishby & Zaslavsky, 2015). These works use tools such as entropy and mutual information to analyze the information transport between NN layers and the information change during training. While such approaches provide an interpretable way to understand the learning process and its limitations, they cannot be directly used to analyze a pre-trained model and the corresponding data paths.

To identify NN data flows, we propose to employ the notion of graph curvature, in particular Ollivier-Ricci Curvature (ORC), as introduced by Ollivier (2009), which provides an effective way to quantify the importance of an edge in the graph. The ORC of a given edge is an indication of the local graph connectivity – edges with positive ORC are typically parts of a strongly connected component, whereas edges with negative ORC normally serve as bridges between connected components. The ORC has been effectively applied to a number of domains that can be modeled as graphs, e.g., road traffic analysis (Wang et al., 2022), internet routing (Ni et al., 2015), biological networks (Znaidi et al., 2023; Sia et al., 2022; 2019), and deep learning (Znaidi et al., 2023), including graph NNs (Liu et al., 2023). We aim to demonstrate that a similar approach can be used for feedforward NNs used for classification purposes.

Inspired by a recent paper, which demonstrates the utility of graph curvature in analyzing NN robustness (Tan et al., 2024), in this paper, we aim to develop a general graph-curvature-based method to identify the main NN data flows and prune the irrelevant ones. We propose a novel neural curvature (NC) metric that can be used to rank NN connections according to their importance, and thus, identify the main data flows. In particular, given a trained NN and a calibration set, we 1) construct a neural graph induced by the NN structure; 2) propose a NC metric to quantify the importance of each NN connection for a given example; 3) rank NN connections according to their NC as observed in the calibration set. Once the ranking is obtained, we perform pruning through removing the least important NN parameters first.

We evaluate the proposed method on small-to-medium size models trained on three image classification datasets, namely MNIST, CIFAR-10 and CIFAR-100. We compare our method with a range of pruning approaches, including magnitude-based pruning (Han et al., 2015), gradient-based pruning (SNIP (Lee et al., 2018) and SynFlow (Tanaka et al., 2020)), and Hessian matrix-based pruning (SparseGPT (Frantar & Alistarh, 2023)), to demonstrate the effectiveness of NC. The magnitude-based method is very effective for small models and heavily regularized models but it often suffers from layer collapse, i.e., removing an entire layer, on larger models where weight magnitudes may differ significantly across layers. While SynFlow and SNIP are able to avoid layer collapse, ultimately they are still constrained by their limited ability to capture edge importance across layers. SparseGPT achieves strong performance on some models, however, it is very susceptible to training hyper-parameters. In contrast, our method provides a continuous measure of edge importance, and is, thus, able to rank edges across the entire model in a robust manner that avoids layer collapse and is better able to identify the main NN data flows. Finally, as the proposed method is computationally expensive, we demonstrate that a hybrid approach, e.g., using magnitude pruning to pre-prune small and unimportant weights, can bring substantial improvements in scalability without sacrificing our method's power in identifying the main data flows.

In summary, this paper makes three contributions: 1) we introduce the notion of neural curvature for NN pruning through analyzing NN data flows; 2) we provide a ranking algorithm for the importance of NN connections, based on neural curvature; 3) we evaluate our method in three image classification datasets so as to demonstrate the effectiveness of the proposed method.

## 2    Related Work

**NN Pruning Methods.** Some studies propose that useless weights can be detected at the initialization stage, in which case the model is pruned before training. Lee et al. (2018); Tanaka et al. (2020); Wang et al. (2020a) are some of the seminal works on pruning at initialization, which are weight-sensitivity-based and provide a weight sensitivity score based on the gradient of the loss and weight. Although these methods are a good first step, locating the important part of the full NN at initialization is still a challenging problem, as also demonstrated in the theoretical analysis, via a mutual information view, by Kumar et al. (2024). Another big group of pruning methods is pruning after training. Weight magnitude pruning was first proposed by Han et al. (2015). They show that weights with large magnitudes are more important and can contain more useful information than smaller weights. The magnitude-based approach has been applied effectively to either structured or unstructured pruning, and also works for LLMs (Frankle & Carbin, 2018; Sun et al., 2023; Dery et al., 2024). There are also studies based on the gradient. Zhao et al. (2019) is weight-sensitivity-based and designed for convolutional NNs (CNNs). Santacroce et al. (2023) proposes a sensitivity and uniqueness-based method for decode-only language models. Frantar & Alistarh (2023) utilizes the Hessian matrix score to prune the unimportant weights for the post-training LLMs. However, many existing post-training pruning methods are developed for fine-tuning and are often architecture-specific. In this paper, we aim to identify the critical information paths in a pre-trained model, which enables a unified and architecture-agnostic post-training pruning approach.

**NN Information Flow Analysis.** Information bottleneck is a famous tool (Tishby et al., 2000; Tishby & Zaslavsky, 2015; Shwartz-Ziv & Tishby, 2017) in NN information flow analysis. It treats a NN as a Markov chain and uses mutual information to analyze the change in information across layers. Based on the information bottleneck principle, Khadivi et al. (2016) study how the entropy of information changes between consecutive layers and develop an optimization problem that can be used for a feedforward NN. Chaddad et al. (2017) and Nasiri (2020) utilize conditional entropy to analyze the information flow in a trained CNN, which can help improve the original CNN's accuracy. Achille et al. (2019) analyzes the information encoded in a trained NN's weights and how it affects its future performance on the test set based on Shannon and Fisher information theory. Wang et al. (2018; 2020b) identify the critical subnet of a NN by directly finding nodes that are important for NN performance, and understand how the NN makes the decision. These methods give an explicit interpretation for NNs' decision behavior; however, they cannot be directly used to identify the main information flows in a trained model. In this paper, we use graph curvature as a local analysis tool that can quantify the importance of each edge in terms of propagating information to the next layer.

**Applications of Graph Curvature Methods.** The ORC (Ollivier, 2007; 2009) is the discrete-space extension of the standard Ricci curvature (Bochner, 1946), which is defined over continuous metric spaces. The ORC is a useful tool to analyze local graph connectivity and identify communities (Sia et al., 2019) in network science. The ORC has also been widely used in different graph-based domains as follows. Gao et al. (2019b) shows that ORC can help locate the vulnerable parts in the road network systems. Wang et al. (2022) utilizes ORC to improve transit network design by analyzing the mismatch of travel demand and supply. Ni et al. (2015) and Salamatian et al. (2022) apply the ORC for internet topologies and network path connectivity analysis. Ricci curvature has also shown effectiveness in biological networks robustness analysis (Simhal et al., 2025). Recently, Ricci curvature has been applied to deep learning as well, including supervising the phase transitions within the dynamics of a time-varying complex network (Znaidi et al., 2023) as well as integrating graph curvature into graph NN design (Wang et al., 2021; Sun et al., 2022; Liu et al., 2023).

## 3    Problem Statement

This section formalizes the problem considered in this paper. We focus on *trained* NN classifiers of the kind $f_\theta : \mathcal{X} \to \mathcal{Y}$, where $\theta$ are the NN parameters, $\mathcal{X} \subseteq \mathbb{R}^{n_x}$ is the input space, and $\mathcal{Y} \subseteq \mathbb{R}^{n_y}$ is the output space. In particular, we consider feedforward NNs, i.e., fully-connected NNs and CNNs, parameterized by $\theta = \{(W_1, b_1), \ldots, (W_L, b_L)\}$, where $L$ is the total number of layers, and $W_i$ and $b_i$ are the weights and biases of layer $i$. We assume $f_\theta$ has either ReLU or Tanh activations after each hidden layer.

Given a trained NN $f_\theta$ and a calibration set $\mathcal{D}$, the problem considered in this paper is to prune the parameters $\theta$ while minimizing the drop in classification accuracy relative to $f_\theta$. To achieve this goal, we aim to design a graph-curvature-

based method to quantify each parameter's importance for the overall NN accuracy and prune parameters according to their importance, starting with the least important.

**Notation.** A weighted directed graph $G(V, E, w)$ is defined as a set of vertices $V = \{v_1, \ldots, v_N\}$ and a set of directed edges $E \subseteq \{(u, v) \mid u, v \in V\}$, where each edge has a positive weight given by $w : E \to \mathbb{R}_+$. For a directed edge $(u, v)$, $u$ and $v$ denote the source and target vertices, respectively. We denote by $\Gamma(v)$ the neighborhood of a vertex $v \in V$; $\Gamma^{in}(v)$ is the incoming neighborhood and $\Gamma^{out}(v)$ the outgoing neighborhood. For a NN $f_\theta$ and an input example $x$, we use $l_{i,j}(x)$ and $n_{i,j}(x)$ to denote the logit and activation of neuron $i$ in layer $j$, i.e., $n_{i,j}(x) = \sigma(l_{i,j}(x))$ for $\sigma \in \{Tanh, ReLU\}$.[1] We use $K_i$ to denote the number of neurons in layer $i$, whereas $[W_{k+1}]_{ij}$ denotes the weight of the NN edge connecting $n_{i,k}$ to $n_{j,k+1}$. We use $[n]$ to denote the list of integers from 1 to $n$.

## 4 Background

This section presents the definition of ORC and its modified version, $\alpha$-Ricci Curvature, as well as their applications. The concept of Ricci curvature (Bochner, 1946) was first introduced in Riemannian geometry to capture the degree to which a given space deviates from being flat (with Euclidean space being the standard example of a flat space). Positive curvature indicates an elevated, spherical shape, whereas negative curvature corresponds to hyperbolic shapes (with a "valley" in between two "hills"). Thus, negative curvature between two points indicates a "bottleneck" such that all shortest paths within a small enough radius include that bottleneck. ORC (Ollivier, 2009) is defined on discrete metric spaces equipped with a Markov chain and can thus be naturally extended to graphs. It aims to capture the notion of a bottleneck edge, as defined below. Further extensions of ORC have been developed, e.g., $\alpha$-Ricci Curvature (Lin et al., 2011), that aim to provide a better graph approximation of the original (continuous) version of Ricci curvature.

**Remark 1** *ORC is most commonly applied to undirected graphs in network science, but several works have extended the definition to directed and weighted graphs (Yamada, 2016; Samal et al., 2018; Eidi & Jost, 2020). In the rest of this paper, we will focus on directed and weighted graphs. We follow a similar mechanism when applying ORC to a directed graph as Eidi & Jost (2020).*

We begin with a graph-based definition of Wasserstein distance, as the building block of ORC.

**Definition 1 (Wasserstein Distance (Optimal Transport Distance))** *Given a directed and weighted graph $G(V, E, w)$, let $u$ and $v$ be the source and target nodes of edge $(u, v)$. Suppose $m_u$ and $m_v$ are two probability distributions supported on the incoming-edge neighborhoods of $u$ and outgoing-edge neighborhoods of $v$, respectively. The Wasserstein distance[2] $W(m_u, m_v)$ between distributions $m_u$ and $m_v$ is given by*

$$W(m_u, m_v) = \inf_{M_{u,v} \in \Pi_{u,v}} \sum_{(u',v') \in \Gamma^{in}(u) \times \Gamma^{out}(v)} d(u', v') M_{u,v}(u', v'), \tag{1}$$

*where $\Pi_{u,v}$ is the set of probability measures $M_{u,v}$ that capture all possible ways of transporting mass from $m_u$ to $m_v$, i.e.,*

$$\sum_{v' \in \Gamma^{out}(v)} M_{u,v}(u', v') = m_u(u'), \qquad \sum_{u' \in \Gamma^{in}(u)} M_{u,v}(u', v') = m_v(v').$$

*In Equation 1, $d(u', v')$ denotes the cost from $u'$ to $v'$, e.g., $d(u, v) = w(u, v)$.*

To better distinguish differences among vertices while ensuring a valid probability measure on each neighborhood, we adopt an exponential normalization, which is one of the standard normalizations in the related work (Ni et al., 2019). This transformation amplifies the contrast between vertices with different measures and guarantees positivity and proper normalization, which are required for the subsequent optimal transport computation.

---

[1]To obtain a neuron ordering in a convolutional layer, we unroll convolution and treat convolutional layers as sparse fully-connected layers.

[2]Typically, the Wasserstein distance with 1-finite moments is written $W_1$ – we drop the subscript to disambiguate from the NN weights notation.

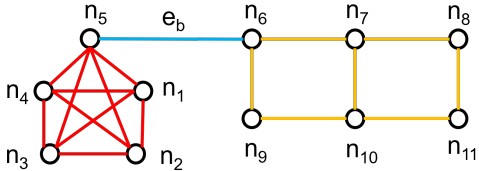

Figure 1: Illustration of Ollivier–Ricci curvature (ORC) on an undirected graph with unit edge weights. Red edges (left) have positive curvature, the blue bridge edge has negative curvature, and yellow edges (right) have zero curvature. ORC is positive for edges within a strongly connected component, negative for edges that serve as bridges between connected components due to higher transport cost, and ORC is zero for edges in plane-like components.

**Definition 2** *(Exponential Neighbor Distribution (Ni et al., 2019)) Given a directed graph $G(V, E, w)$, the probability distribution $m_x$ over the neighborhood of a node $x$ is defined via exponential normalization:*

$$m_x(u) = \frac{\exp\big(-[h_x(u)]^2\big)}{\sum\limits_{u_i \in \bar{\Gamma}(x)} \exp\big(-[h_x(u_i)]^2\big)}. \tag{2}$$

*If $x$ is a source node, $\bar{\Gamma}(x) := \Gamma^{in}(x)$; if $x$ is a target node, $\bar{\Gamma}(x) := \Gamma^{out}(x)$. Here, for an incoming neighbor $u$, $h_x(u) = w(u, x)$ denotes the value function of node $u$ (in the case of outgoing neighbor, $h_x(u) = w(x, u)$).*

**Definition 3 (ORC (Ollivier, 2009))** *Given a directed and weighted graph $G(V, E, w)$, the ORC along a directed edge $(u, v)$ is defined as*

$$\kappa_{ORC}(u, v) = 1 - \frac{W(m_u, m_v)}{d(u, v)}. \tag{3}$$

**Example 1** *Consider the example undirected graph in Figure 1 where all edges have weight 1, therefore each neighbor is assigned the same probability. We will illustrate how to calculate the ORC of the bridge edge, $e_b$, connecting nodes $n_5$ and $n_6$. To calculate the Wasserstein distance, we first need the distributions $m_{n_5}$ and $m_{n_6}$, which assign probabilities to neighbors of $n_5$ and $n_6$, respectively, depending on their weights. Since all edges have the same weight, we have $m_{n_5} = [0.2\ 0.2\ 0.2\ 0.2\ 0\ 0.2\ 0\ 0\ 0\ 0\ 0]$ and $m_{n_6} = [0\ 0\ 0\ 0\ 1/3\ 0\ 1/3\ 0\ 1/3\ 0\ 0]$. The optimal transport cost from $m_{n_5}$ to $m_{n_6}$ is $0.2 + 3*(2/3 - 0.2) + 1*(1/3)$, i.e., $W(m_{n_5}, m_{n_6}) \approx 1.93$, hence $\kappa_{ORC}(e_b) \approx -0.93$.*

The standard ORC formulation assumes that the probability measure $m_u$ around a node $u$ is induced by a simple random walk and that the underlying metric assigns uniform distances between adjacent nodes. These assumptions are often not true in weighted graphs, where edge weights may vary a lot, leading to a more discrete and non-uniform neighborhood geometry and potentially large variations in local distances. As a result, a direct application of the standard ORC definition may fail to accurately capture the geometric structure of weighted graphs. To address this limitation, Lin et al. (2011) modified Definition 3 and proposed $\alpha$-Ricci Curvature, denoted by $\kappa_\alpha(u, v)$, where $\alpha$ is a parameter that allows additional probability mass to remain at the source and target nodes, thereby better approximating a continuous neighborhood structure on weighted graphs.

**Definition 4 ($\alpha$-ORC (Lin et al., 2011))** *Given a weighted and directed graph $G(V, E, w)$, for any $\alpha \in [0, 1]$, the probability measure $m_x$ over the neighborhood of a node $x$ is modified to $m_x^\alpha$, which assigns an $\alpha$-fraction of the probability mass to $x$ itself and distributes the remaining mass among its neighbors. The definition is given by:*

$$m_x^\alpha(n) = \begin{cases} \alpha, & \text{if } n = x, \\ (1 - \alpha)m_x(n), & \text{if } n \in \bar{\Gamma}(x), \\ 0, & \text{otherwise}, \end{cases} \tag{4}$$

*where $m_x(n)$ and $\bar{\Gamma}(x)$ are defined in Definition 2.*

*Then the $\alpha$-ORC curvature, $\kappa_\alpha(u,v)$, between two vertices $u$ and $v$ is given by*

$$\kappa_\alpha(u,v) = 1 - \frac{W(m_u^\alpha, m_v^\alpha)}{d(u,v)}. \tag{5}$$

Note that $\kappa_{ORC}(u,v) = \kappa_0(u,v)$ and $\kappa_1(u,v) = 0$, since $W(m_u^1, m_v^1) = d(u,v)$. Thus, it is not clear what value of $\alpha$ is most suitable in Definition 4. To address this issue, Lin et al. (2011) further define $h(\alpha) = \kappa_\alpha(u,v)/(1-\alpha)$ and prove that $h(\alpha)$ is an increasing function and remains bounded as $\alpha \to 1$. This implies that the limit

$$lim_{\alpha \to 1} \frac{\kappa_\alpha(u,v)}{1-\alpha}$$

exists. This limit is denoted by $\kappa(u,v)$, the Ricci curvature on graphs.

**Definition 5 (Ricci Curvature on Graphs (Lin et al., 2011))** *Given a directed graph $G(V, E, w)$, the graph Ricci curvature of a directed edge $(u, v)$ is defined as:*

$$\kappa(u,v) = lim_{\alpha \to 1} \frac{\kappa_\alpha(u,v)}{1-\alpha}. \tag{6}$$

**Remark 2** *In this paper, we build on Definition 5 for the proposed concept of neural Ricci curvature.*

# 5 Approach

This section describes the proposed approach of using graph curvature to rank NN edges according to their importance. As discussed in the introduction, graph curvature has been used to analyze a number of domains that can be modeled as graphs. For example, in road and transit traffic analysis (Gao et al., 2019a; Wang et al., 2022), researchers used graph curvature to identify bottleneck segments which could be upgraded or which require better alternatives. Analyzing NNs has clear analogies and differences with road analysis. In particular, NN edges can be considered as roads that transport data, with NN weights determining the capacity of each road. Thus, it is natural to focus on bottleneck NN edges as potentially important for the overall NN functionality, similar to the traffic analysis case. The main challenge is how to encode the NN as a weighted graph, specifically how to handle 1) activation functions and 2) changing data patterns depending on the input example.

## 5.1 Overview

A high-level overview of the proposed approach is shown in Figure 2 and summarized in Algorithm 1, which is provided in Appendix A.5. Given a trained NN $f_\theta$ and a calibration set $\mathcal{D}$, for each NN connection $e$ we aim to provide a curvature value that quantifies this edge's importance. Similar to the standard ORC setting, edges with lower curvature are considered more important for the overall NN functionality. We quantify this process through the following three steps: 1) create a neural graph $G_{f_\theta}$, as introduced in prior work (Tan et al., 2024; Xiao et al., 2024); 2) for each example $x$ and each NN connection $e$, calculate the neural curvature of $e$, as described in this section; 3) for each NN connection $e$, store the minimum curvature value of $e$ over all the examples. Since more negative curvature is associated with a stronger contribution to information flow and model performance, the minimum curvature captures the most influential role that the connection may play across the dataset. The rest of this section formalizes the notions of a neural graph as well as the novel definition of neural curvature.

## 5.2 Neural Graph and Neural Curvature

This subsection introduces the concept of neural curvature. We begin with a definition of a neural graph, i.e., a weighted graph induced by the NN architecture, as introduced by prior work (Tan et al., 2024; Xiao et al., 2025).

**Definition 6 (Neural Graph (Tan et al., 2024; Xiao et al., 2025))** *Consider a NN $f_\theta$ with $L$ layers, where $V_l = \{v_{1,l}, \ldots, v_{K_l,l}\}$ denotes the neuron set of layer $l$ and $E_{l,l+1} = \{(v_{1,l}, v_{1,l+1}), \ldots, (v_{K_l,l}, v_{K_{l+1},l+1})\}$ denotes*

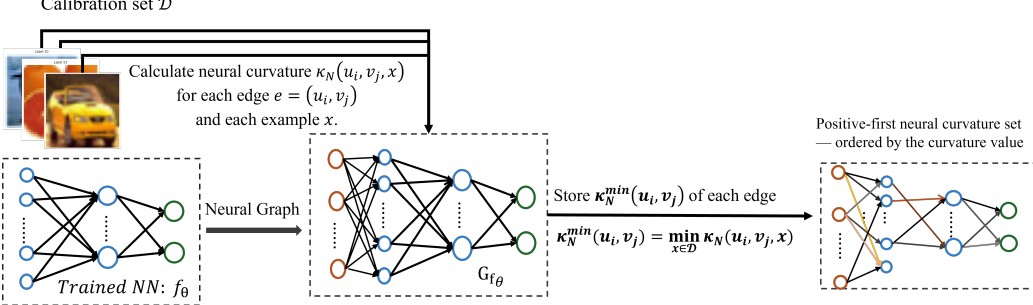

Figure 2: Approach overview. Given a trained NN $f_\theta$ and a calibration set $\mathcal{D}$, we aim to identify the importance of NN connections, respectively. Specifically, we 1) construct a neural graph $G_{f_\theta}$ induced by the NN architecture (including an input layer); 2) for each edge $(u, v)$ in the neural graph and each example $x \in \mathcal{D}$, we calculate the neural curvature $\kappa_N(u, v, x)$, as defined in Definition 9; 3) we order the whole edge set by positive-first-curvature edges.

*the NN connections from layer $l$ to $l + 1$. A neural graph $G_{f_\theta} = (V_{f_\theta}, E_{f_\theta}, w_{f_\theta})$ is defined as follows:*

$$V_{f_\theta} = V_0 \cup V_1 \cup V_2 \cup \cdots \cup V_L$$
$$E_{f_\theta} = E_{0,1} \cup E_{1,2} \cup \cdots \cup E_{L-1,L}$$
$$w_{f_\theta}(v_{i,l}, v_{j,l+1}) = \frac{1}{|[W_{l+1}]_{ij}|},$$

*where $V_0 = \{v_{1,0}, \ldots, v_{n,0}\}$ adds one vertex per input dimension (i.e., the NN input layer). Thus, $G_{f_\theta}$ has one vertex per input dimension, one vertex per neuron in the NN and one edge per NN connection. The weight of each edge in the graph is the inverse of the magnitude of the corresponding NN weight.[3]*

Definition 6 describes the graph induced by the NN structure. Note that setting the graph weights to be the inverse of the NN weights signifies that the "cost" of sending data through an edge is inversely proportional to the NN weight; intuitively, the larger the weight, the more data is sent. Other weightings will be considered in future work as well.

Note that Definition 6 does not depend on the input example and as such cannot be easily used to capture the NN information flow. We incorporate the input example in the curvature calculation as follows. First note that the standard $\alpha$-Ricci curvature in Definition 4 derives the neighbor distributions, $m_u$ and $m_v$, directly from the graph weights. Unlike the standard definition, which is static, we incorporate the input examples into the curvature calculation. Specifically, we modify the neighbor distribution depending on the magnitude of each neuron activation for a given input example. By incorporating these node values, the curvature captures how the input propagates through the NN, making it data-dependent and providing a more meaningful measure of the NN's behavior for a specific example.

**Definition 7 (Neural Neighbor Distribution)** *Consider a neural graph $G_{f_\theta}$ and a corresponding vertex $v_{i,l}$ for $l \geq 1$, and a given input example $x$. The incoming neural neighbor distribution of $v_{i,l}$ is defined as follows:*

$$\mu_{v_{i,l}}(v_{j,l-1}, x) = \frac{\exp\left(-[h_n(v_{j,l-1}, x)]^2\right)}{\sum\limits_{v_{k,l-1} \in \Gamma^{in}(v_{i,l})} \exp(-[h_n(v_{k,l-1}, x)]^2)},$$
$$h_n(v_{j,l-1}, x) = \frac{1}{n_{j,l-1}^{norm}(x)}$$

*where $\mu_{v_{i,l}}(v_{j,l-1}, x)$ is defined analogously to Definition 2. Here, $h_n(v_{j,l-1}, x)$ is the value function of node $v_{j,l-1}$, which depends on the neural activation values with input $x$, and $n_{j,l-1}^{norm}(x)$ is a normalization function applied in two steps as follows:*

*1. Let $n_{min} = \min\{|n_{1,l-1}(x))|, \ldots, |n_{K_{l-1},l-1}(x)|\}$ and $n_{max} = \max\{|n_{1,l-1}(x))|, \ldots, |n_{K_{l-1},l-1}(x)|\}$.*

---

[3]Although prior work Tan et al. (2024) considers an additional normalization of the graph weights depending on their sign, we found it to be unnecessary.

2. **Per-layer neuron normalization:** $n_{j,l-1}^{norm}(x) = \frac{|n_{j,l-1}(x)| - n_{min}}{n_{max} - n_{min}}$;

*The outgoing neighbor distribution is defined similarly, but using the values in layer $l+1$. If $v_{i,l}$ is in the input layer, then the input neighbor distribution is just a mass of 1 on $v_{i,l}$ itself. Finally, if $v_{i,l}$ is in the output layer, then the outgoing neighbor distribution is just a mass of 1 on $v_{i,l}$.*

Intuitively, Definition 7 captures the fact that the neighbor distribution is determined by the value of each neuron. The inversion in $V_n(v_{j,l-1}, x)$ ensures that larger neuron values get a weight that is closer to 0. In turn, this ensures large neurons get a larger density after being sent through a standard normal distribution in $\mu_{v_{i,l}}(v_{j,l-1}, x)$, as proposed in prior work on normalizing neighbor distributions (Ni et al., 2019). Finally, the normalization in Step 2 just ensures that all neuron values are normalized within the range $(0, 1]^4$, to prevent numeric issues caused by ReLU, which can have large positive values. We emphasize again that, unlike the original neighbor distribution in Definition 2, which is calculated based on the weights connecting those neighbors, Definition 7 weights neighbors according to their corresponding neuron (activation) values.

To further capture the effect of activation functions on an edge cost, we note that some NN connections' importance may change depending on the activations of the neurons they go in/out of. In particular, connections that go into a saturated neuron (in the case of Tanh) or in/out of a non-activated neuron (in the case of ReLU) have reduced importance for the NN's output on the current example. In such cases, the effective edge cost increases – this is captured through rescaling the edge cost by an additional term that quantifies the effect of the non-linearity on the given edge. Thus, we define an edge's neural cost as follows.

**Definition 8 (Neural Edge Cost.)** *Consider a directed edge $(v_{i,l}, v_{j,l+1})$ between layers $l$ and $l+1$ in a neural graph. Let $\sigma \in \{Tanh, ReLU\}$ be the activation function in the corresponding NN. The* neural cost *of $(v_{i,l}, v_{j,l+1})$ is defined as follows:*

$$d_{ReLU}(v_{i,l}, v_{j,l+1}, x) = \frac{w_{f_\theta}(v_{i,l}, v_{j,l+1})}{\min\{\beta(v_{i,l}(x)), \beta(v_{j,l+1}(x))\}}$$

$$d_{Tanh}(v_{i,l}, v_{j,l+1}, x) = \frac{w_{f_\theta}(v_{i,l}, v_{j,l+1})}{\beta(v_{j,l+1}(x))}.$$

*where $\beta(v) = \sigma(v)/v$ is the fraction of information that passes through the activation function. If $\beta(v) = 0$, then the cost is set to $\infty$.*

**Remark 3** *In the case of ReLU, $\beta(v) \in \{0, 1\}$. In the case of Tanh, $\beta(v) \in (0, 1]$, where we define $\beta(0) = 1$ in the case of $Tanh$ since $\lim_{v \to 0} Tanh(v)/v = 1$.*

Intuitively, Definition 8 increases an edge's cost based on the fraction of data that is removed by the activation function, which we use as a proxy for the reduced importance of the corresponding NN connection. Note that in the case of ReLU, we consider both the source and target neurons – in either case, we set the edge's cost to $\infty$ if the ReLU is not activated. In the case of Tanh, we only consider the target neuron since even if the source neuron is saturated, that connection may still carry information.

**Remark 4** *Note that the neural edge cost does not update the weights in the neural graph $G_{f_\theta}$, i.e., the Wasserstein distance is based purely on the neural neighbor distribution in Definition 7 and the weight function in Definition 6. The neural edge cost is used only in the neural curvature calculation in the following definition.*

We are now ready to state the definition of neural curvature, which is the main technical contribution of this paper.

**Definition 9 (Neural Curvature)** *Consider a trained NN $f_\theta$, an input example $x$, and the corresponding neural graph $G_{f_\theta}(V_{f_\theta}, E_{f_\theta}, w_{f_\theta})$. The* neural curvature *of an edge $(v_{i,l}, v_{j,l+1}) \in E_{f_\theta}$ is defined as*

$$\kappa_N(v_{i,l}, v_{j,l+1}, x) = \lim_{\alpha \to 1} \frac{\kappa_{\alpha,N}(v_{i,l}, v_{j,l+1}, x)}{1 - \alpha}, \tag{7}$$

---

[4]Note that a value of zero occurs only when $|n_{j,l-1}(x)| = n_{min}$; the lower bound of the interval is effectively open to prevent division by zero – we achieve this in the implementation by adding a small positive constant.

*where the $\alpha$-neural-curvature is defined as:*

$$\kappa_{\alpha,N}(v_{i,l}, v_{j,l+1}, x) = 1 - \frac{W\left(\mu^{\alpha}_{v_{i,l}}(\cdot, x), \mu^{\alpha}_{v_{j,l+1}}(\cdot, x)\right)}{d_{\sigma}(v_{i,l}, v_{j,l+1}, x)}. \tag{8}$$

*In equation 8, $\sigma \in \{Tanh, ReLU\}$; note that $\mu^{\alpha}_{v_{i,l}}$ and $\mu^{\alpha}_{v_{j,l+1}}$ are calculated according to Definition 7 and equation 4. Finally, if $d_{\sigma}(v_{i,l}, v_{j,l+1}, x) = \infty$, then $\kappa_{\alpha,N}(v_{i,l}, v_{j,l+1}, x) = 1$ if $l = 0$ or $l = L - 1$, and $\kappa_{\alpha,N}(v_{i,l}, v_{j,l+1}, x) = 2$, otherwise, as shown in Proposition 1 next.*

Note that $\beta(v)$ is often 0 in the case of ReLU activations, which in turn causes $d_{\sigma}$ to be set to $\infty$ in Definition 9. It turns out that, as the cost gets large, the neural curvature converges to either 1 or 2, depending on the edge's position in the graph, as shown next (the proof is provided in Appendix A.6).

**Proposition 1 (Neural Curvature with Infinite Costs)** *Consider a neural graph $G_{f_{\theta}}$ and consider the neural curvature $\kappa_N(v_{i,l}, v_{j,l+1}, x)$ for a directed edge $(v_{i,l}, v_{j,l+1})$. When the edge cost gets large, i.e., $d_{\sigma}(v_{i,l}, v_{j,l+1}, x) \to \infty$, the neural curvature converges as follows:*

$$\lim_{d_{\sigma}(v_{i,l}, v_{j,l+1}, x) \to \infty} \kappa_N(v_{i,l}, v_{j,l+1}, x) = \begin{cases} 1, & \text{if } l = 0 \text{ or } l = L - 1, \text{ i.e., } (v_{i,l}, v_{j,l+1}) \text{ connects an input/output layer} \\ 2, & \text{otherwise.} \end{cases}$$

### 5.3 Treatment of Convolutional Parameters

Note that each NN convolutional parameter results in multiple edges in the corresponding neural graph. While our neural curvature framework can, in principle, compute a separate curvature value for each edge in a convolutional layer (i.e., each connection between input and output feature maps at specific spatial locations), we adopt a simplified approach so as to rank NN parameters as opposed to resulting edges in the neural graph.

Specifically, for each convolutional parameter, we take the minimum curvature across all edges corresponding to a given parameter, capturing the strongest contribution that the parameter can make to information propagation. This approach allows us to assign a single curvature value to each NN parameter such that we can compare convolutional and fully-connected parameters.

### 5.4 NN Connection Ranking

Given Definition 9, we rank NN connections according to the minimum neural curvature value that each edge attains across all examples in the dataset. We take the minimum over samples because more negative curvature corresponds to stronger information flow and greater functional importance of an edge. Specifically, we combine all the edges and construct the curvature set as follows:

$$\mathcal{C}_{\text{full}} = \left\{ \min_{x \in \mathcal{D}} \min_{(u,v) \in \mathcal{E}(p)} \kappa_N(u, v, x) \,\Big|\, p \in \theta \right\},$$

where $\mathcal{E}(p)$ denotes all edges in the neural graph that correspond to NN parameter $p$. We interpret the curvature values such that edges with the most negative curvature are considered the most important for information propagation, while edges with the most positive curvature are considered the least important. Accordingly, $\mathcal{C}_{full}$ can be ranked from highest to lowest curvature to prioritize the least critical connections, as summarized in Algorithm 1.

In Section 6, we evaluate the importance of the proposed ranking algorithm and provide strong evidence that negative-curvature edges carry the essential NN data flow, whereas the majority of positive edges have no impact on NN performance and can be pruned at comparatively lower cost in accuracy.

### 5.5 Neural Curvature Implementation

The overall implementation of neural curvature is based on the `GraphRicciCurvature` library (Ni et al., 2019), which efficiently handles graph construction and curvature computations on the CPU using Python. To better integrate

with PyTorch and leverage GPU acceleration, which is widely used in NN training, we reimplemented parts of the library in PyTorch, enabling curvature computations directly on the GPU. Specifically, while the Wasserstein distance computation still needs to be performed on the CPU (effectively solving a linear program for each edge), the shortest-path calculation can be performed on the GPU, through a dynamic programming algorithm as follows.

Since the neural graph $G_{f_\theta}$ is directed and has a layer-by-layer structure, we can apply dynamic programming to compute the shortest-path distances required for the Wasserstein distance in the curvature calculation. For a neural graph $G_{f_\theta}$, we define an adjacency cost matrix of layer $l$ as a matrix $C^{(l)}$, where each entry $C_{ij}^{(l)} = w_{f_\theta}(v_{i,l-1}, v_{j,l})$ is the cost associated with the edge from neuron $v_{i,l-1}$ to neuron $v_{j,l}$.

Let $D^{(k,l)}$ denote the distance matrix of minimum accumulated path cost between neurons in layer $k$ and neurons in layer $l$, for any $k < l$. $D^{(k,l)}$ is computed via dynamic programming recursion:

$$D^{(k,k+1)} = C^{(k+1)}, \quad l = k+1,$$
$$D^{(k,l)} = C^{(k+1)} \oplus D^{(k+1,l)}, \quad l > k+1.$$

where $\oplus$ is defined as $(A \oplus B)_{ij} = \min_m (A_{im} + B_{mj})$. This procedure can be efficiently implemented on a GPU, thereby allowing us to scale our approach to large CNNs.

Finally, to approximate the limit of the neural curvature as Definition 5, we set $\alpha = 0.9$. Based on our empirical observations, the $\alpha$-curvature does not change significantly beyond that point, so it is a good compromise between approximation accuracy and numerical stability.

# 6 Experiments

This section provides an experimental evaluation of the proposed ranking algorithm for NN connections. We aim to show that our approach can effectively identify both the least and the most important connections in a given NN. We evaluate the proposed approach on three image classification tasks, namely MNIST (LeCun et al., 1998), CIFAR-10, and CIFAR-100 (Krizhevsky & Hinton, 2009) datasets. All experiments were run on a 96-core machine with an NVIDIA A40 unit. The following section only shows selected model results; the remaining results are discussed in Appendix A.

## 6.1 Experiment Design

Given a trained model $f_\theta$ and a calibration dataset $\mathcal{D}$, the evaluation has the following structure:

- Generate a sorted edge set following Algorithm 1.

- For each model, compute curvature edge sets using a validation subset: 10 examples (one per label) for CIFAR-10, 100 examples (one per label) for CIFAR-100, and 100 examples (10 per label) for MNIST.

- Evaluate the impact of the ordered edge set via a pruning experiment: starting from the full network, progressively remove edges according to the sorted set and measure the corresponding effect on test accuracy.

- For evaluation, we compare our method with standard pruning techniques, including magnitude-based pruning (Han et al., 2015), gradient-based methods (SNIP (Lee et al., 2018), SynFlow (Tanaka et al., 2020)), and the Hessian-based SparseGPT (Frantar & Alistarh, 2023) on pre-trained models without retraining. Although SNIP and SynFlow were originally proposed for pruning at initialization, they can also be applied to trained models. SparseGPT, designed for post-training LLM pruning, supports CNN and linear layers, while magnitude pruning applies to all settings. [5]

- Explore scalability improvements, e.g., through a hybrid approach where i) magnitude-based pruning is used to pre-prune small and unimportant weights and ii) our method is applied on the remaining parameters.

---

[5]We perform 100 iterations for SynFlow. We adopt the implementation in https://github.com/ganguli-lab/Synaptic-Flow (Tanaka et al., 2020) for Magnitude, SNIP, and SynFLow; https://github.com/IST-DASLab/sparsegpt (Frantar & Alistarh, 2023) for SparseGPT.

**Model description.** To demonstrate the effectiveness of our method, we train a variety of NN architectures with three different losses and two activation functions. On MNIST, we use a CNN architecture, based on the LeNet5 architecture (LeCun et al., 1998); on CIFAR-10 and CIFAR-100, we trained a CNN that adopts a VGG-style convolutional architecture (Simonyan & Zisserman, 2015), with the main differences being the removal of padding in convolutional layers, pooling after blocks, and a reduction in fully connected layer size. We refer to this variant as VGG9-lite. In each task, the training set is separated into a 50k training set and a 10k validation set (note that a subset of the validation set is actually used, as per the numbers provided at the beginning of this subsection). Three different training schemes are applied: 1) cross-entropy (CE) loss; 2) cross-entropy loss with weight decay regularization (WD); 3) adversarial training (AT) (Madry et al., 2017), as well as two activation functions: 1) ReLU and 2) Tanh. In total, we have trained six models on MNIST, six models on CIFAR-10, as well as three models on CIFAR-100 (ReLU). All model hyper-parameters are shown in the Appendix A.1. We emphasize that our results are equally effective across all considered combinations.

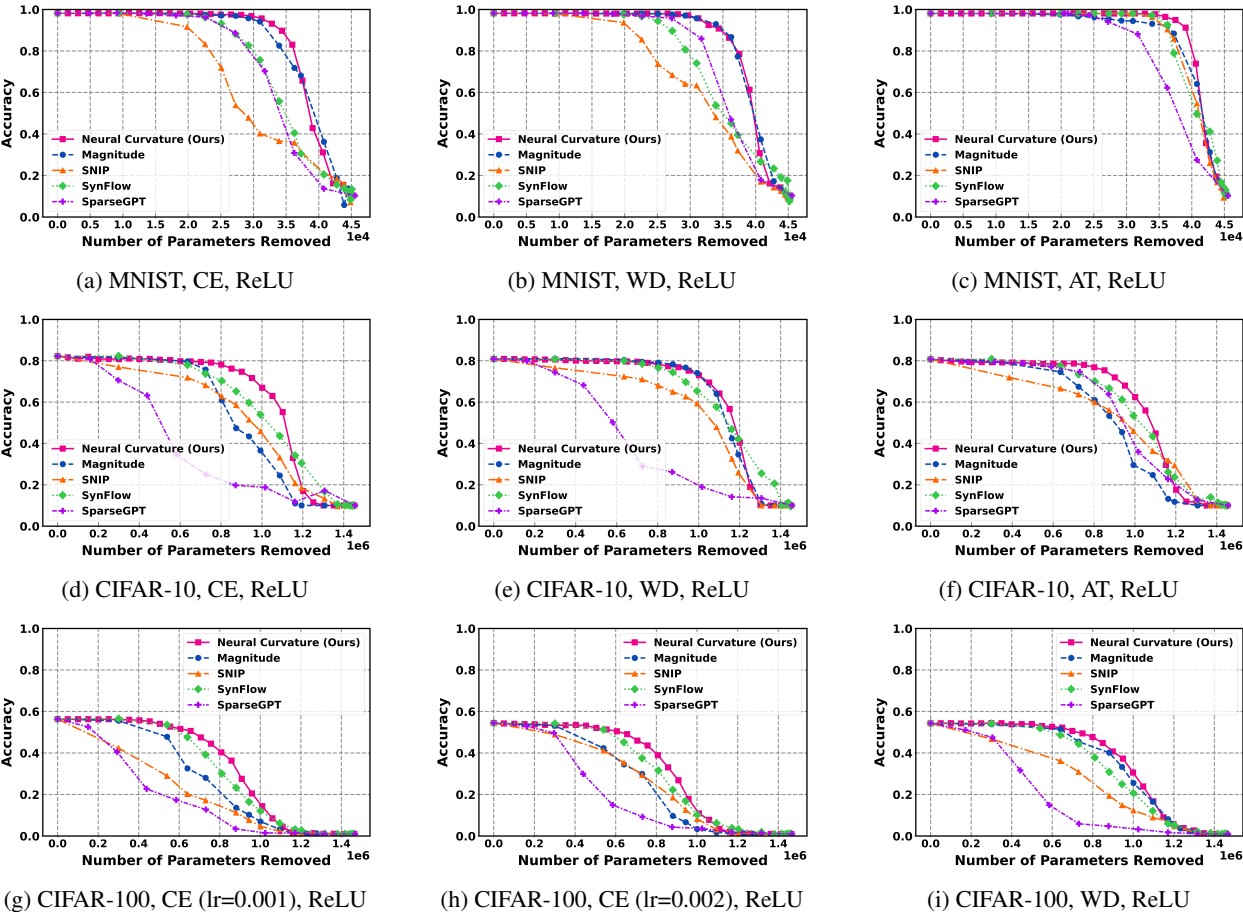

Figure 3: Edge removal evaluation on MNIST, CIFAR-10, and CIFAR-100 (lr means learning rate). Each subfigure shows a comparison of our neural curvature algorithm with existing pruning baselines.

## 6.2 Pruning-Based Comparison

The pruning results for all ReLU models are presented in Figure 3, and we present the results for Tanh models in Appendix A.7. On MNIST, our method performs comparably to magnitude pruning. In this setting, the network size is modest and less over-parameterized, leaving little redundancy to exploit. By comparison, SNIP, SynFlow, and SparseGPT consistently underperform across all model architectures.

On CIFAR-10, although pruning-based methods generally provide a meaningful separation between more and less important connections, the results indicate that our method is substantially more effective across most model configu-

rations, particularly for CE- and AT-trained networks. Magnitude pruning performs well for WD-regularized models, likely because weight decay naturally pushes many parameters toward zero, increasing the ease of identifying redundant weights; nevertheless, our method achieves performance that is superior to magnitude pruning across almost all experimental settings, and a deeper discussion about the effect of the WD hyper-parameter is provided in an ablation study in Section 6.3.4. SNIP behaves similarly to a magnitude-times-gradient (a first-order Taylor approximation) criterion when used on a pre-trained network, tending to eliminate parameters that are effectively inactive but struggling to distinguish among weights that already contribute substantially to predictive performance. SynFlow, by contrast, is data-free and therefore produces pruning dynamics similar to magnitude pruning, yet it consistently initiates pruning in the largest layers. This behavior helps prevent catastrophic layer collapse but may not be effective when layer size is not perfectly correlated with the number of redundant parameters. We also provide more results for SynFlow with different numbers of iterations in Appendix A.10 for better evaluation. SparseGPT was originally evaluated primarily on fully-connected and attention layers in Frantar & Alistarh (2023). Convolutional layers present significantly more challenges due to spatial coupling and the fact that each parameter is multiplied by several input dimensions, which makes it difficult to quantify the parameter's importance. In particular, SparseGPT achieves performance comparable to SynFlow on the AT model, but underperforms on the CE and WD models, indicating that its effectiveness depends on the training objective and is not consistent across different training settings. Overall, our method achieves the best or on-par performance with state-of-the-art approaches across all training configurations and activation function regimes, highlighting its effectiveness and broad applicability.

We additionally report results on CIFAR-100,[6] where our method outperforms all other baselines by a substantial margin, similar to the CIFAR-10 CE model. Although CIFAR-100 is similar to CIFAR-10 in structure, its input examples and associated data pathways are more complex due to the increased number of classes and visual variability. Consequently, despite the training protocol including weight decay, our method remains more effective than magnitude pruning, and continues to consistently outperform SNIP, SynFlow, and SparseGPT. This observation provides further evidence that the algorithm proposed in this work is able to identify the primary data flow paths within NNs.

## 6.3 Ablation Study

For a deeper analysis of the proposed method, we also provide an ablation study where we investigate the effect of the proposed neural modules for neural curvature calculation, the data requirements of our method, the effect of the WD hyperparameter, as well as randomization robustness in the training/validation set. We also provide more detailed exploration, such as the results of per-layer removal, WD hyper-parameter of Tanh models, and the effect of the number of SynFlow iterations, in Appendix A.

### 6.3.1 Effect of Negative-Curvature Edge Removal

While in the previous section we demonstrated that our method is very effective at identifying the least important NN connections as the ones with the highest neural curvature (positive-first pruning), we now also show that negative-curvature edges tend to be the most important. We show the positive-first and negative-first edge removal results for one MNIST, CE model (ReLU), and two CIFAR-10 CE models (ReLU and Tanh). As shown in Figure 4, the results show a very convincing tendency that the negative edges constitute the primary data flow, whereas the majority of the positive edges have no impact on the NN accuracy. In particular, the accuracy degrades sharply when negative edges are removed first, while it remains almost the same when the majority of positive edges are removed first. We emphasize that the results show the same trends across all models and training algorithms – please consult the Appendix A.8 for the full results.

This demonstrates both the effectiveness of our approach as well as the fact that standard models do not utilize the vast majority of the available weights. In addition, the curvature annotations above the pruning curves reveal a consistent trend: model accuracy decreases as the removed edges include those with curvature values closer to zero. This provides further evidence that highly negative-curvature edges are structurally important for maintaining predictive performance. For example, in the CNN model shown in Figure 4a, accuracy remains stable while pruning edges with positive curvature, and begins to noticeably drop only once the removed edges include those with curvature values near or below zero. A similar pattern appears in the VGG9-lite experiments (Figures 4b and 4c), where the curvature value at which accuracy first declines is substantially smaller than the value associated with the initial flat region of the

---

[6]We do not include an AT-trained model for CIFAR-100 since those models incur a significant drop in accuracy.

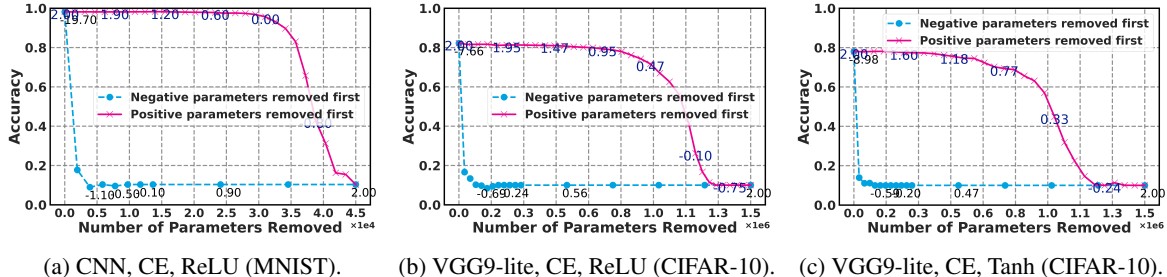

(a) CNN, CE, ReLU (MNIST).  (b) VGG9-lite, CE, ReLU (CIFAR-10).  (c) VGG9-lite, CE, Tanh (CIFAR-10).

Figure 4: Edge removal evaluation on MNIST and CIFAR-10 using our neural curvature algorithm. Each subfigure shows a comparison of the impact of removing edges in order of positive-curvature-first versus negative-curvature-first. Numbers on the curves indicate the minimum curvature per removed edge over the validation set.

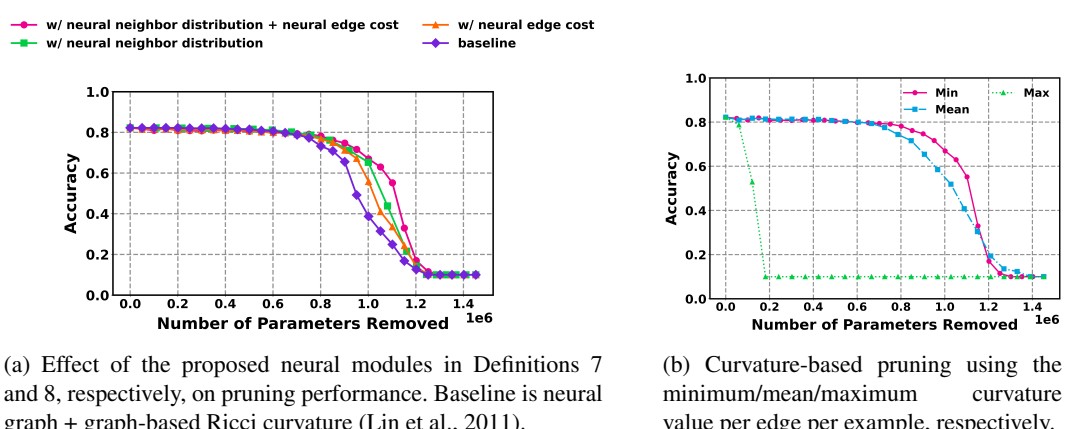

(a) Effect of the proposed neural modules in Definitions 7 and 8, respectively, on pruning performance. Baseline is neural graph + graph-based Ricci curvature (Lin et al., 2011).

(b) Curvature-based pruning using the minimum/mean/maximum curvature value per edge per example, respectively.

Figure 5: Ablation experiment of different neural curvature calculations (VGG9-lite, CE ReLU, on CIFAR-10).

curve. Moreover, as illustrated by the blue curve, pruning edges with strongly negative curvature can abruptly collapse accuracy, effectively breaking the network. These observations indicate that curvature is not merely correlated with edge importance, but serves as a reliable and fine-grained signal for determining which edges should be preserved during pruning.

### 6.3.2 Effect of Neural Modules and Edge Ranking Strategies on Curvature Calculation

Our algorithm for neural curvature computation is largely aligned with the standard formulation Lin et al., 2011, with two modifications. First, we use the neural neighbor distribution as defined in Definition 7. Second, we apply a neural edge cost to factor in the activation function effect as defined in Definition 8. In particular, for ReLU-based models, we assign an infinite cost to a neural edge when either the source or target node has $\beta = 0$. To assess the importance of these two components, we perform an ablation study (Figure 5a). As shown in the results, the baseline Lin et al., 2011, which is static and only uses the weight magnitudes of the NN, is significantly outperformed by either modification based on incorporating the input examples in addition to the NN structure.

We further present an ablation study to compare different strategies to compute edge rankings over all the examples, including using minimum, mean, and maximum value, shown in Figure 5b. As discussed in Section 5.4, compared to using mean or maximum value, aggregating with the minimum value best captures the critical importance of each connection, leading to the strongest overall performance.

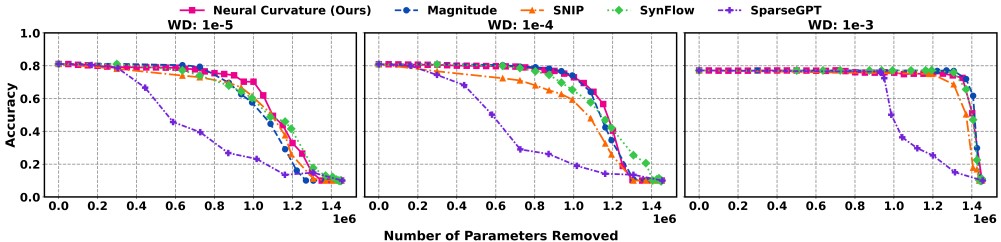

Figure 7: Full model edge removal comparison between magnitude-based and ours (VGG9-lite, WD ReLU, on CIFAR-10). Models are trained with different WD parameters.

### 6.3.3 Data Requirements

While the baselines we compare against (e.g., SynFlow) are data-free, our curvature computation requires data to estimate neuron activations. It is therefore important to assess how much data is needed for our method to produce reliable pruning decisions. Figure 6 provides an illustration of the data requirements of our method, for the same model as in Figure 16. We compare the results with using only one example of label 0, five examples of labels $0 - 4$, 10 examples (one example per label), 20 examples (two examples per label), 50 examples (five examples per label), and 100 examples (10 examples per label). As shown in the figure, one example per label is sufficient to recover most of the pruning behavior observed with the full dataset. While adding more examples per label enables a larger number of parameters to be pruned, it leads to a faster decline near the end. Notably, our method is highly data-efficient and can effectively separate the two sets even with only 10 calibration examples. Increasing the number of examples does not substantially change the results and, in rare cases, may slightly alter the separation, likely due to noise in the activation statistics that perturbs the curvature distribution.

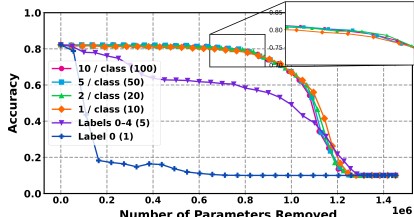

Figure 6: Ablation experiment for edge removal using different numbers of calibration examples (VGG9-lite, CE ReLU, on CIFAR-10).

### 6.3.4 Effect of Weight Decay on Neural Curvature and Magnitude

WD is known to influence the distribution of weights in NNs, which can in turn affect pruning behavior. The results in the previous sections showed that when WD training is used, magnitude-based pruning improves and can approach or even exceed the performance of our method. In this section, we systematically investigate models trained with different WD hyperparameters to examine how WD affects pruning outcomes and to further illustrate its impact on different pruning methods. We compare full-model edge removal across models trained with different weight decay (WD) hyperparameters on CIFAR-10. Figure 7 shows results for VGG9-lite ReLU trained with WD parameters $1e-5$, $1e-4$, and $1e-3$, and the results for Tanh models are presented in Appendix A.9. The results indicate that larger WD values substantially improve the effectiveness of magnitude pruning. Effectively, WD helps eliminate unimportant weights by pushing them toward very small values, thus compressing the valid data paths within the network. As the WD parameter increases, magnitude pruning becomes more efficient; however, our method still achieves comparable performance on ReLU models. Finally, it is important to note that increasing the WD hyperparameter typically results in lower accuracy and requires significantly more training effort. Thus, although it is possible to obtain models which are effectively pruned through magnitude-based pruning, one may need to trade off other model properties, such as accuracy or computation.

### 6.3.5 Statistical Robustness Analysis

To assess statistical robustness, we conducted additional experiments using different random seeds for: 1) calibration samples selection; 2) random model initialization of trainable model, CIFAR-10 models (CE, ReLU). The calibration set used to calculate the curvature is randomly selected from the validation set, and different random seeds are used to guarantee our algorithm is evaluated on different data. Also, we use different seeds to get different model initializations

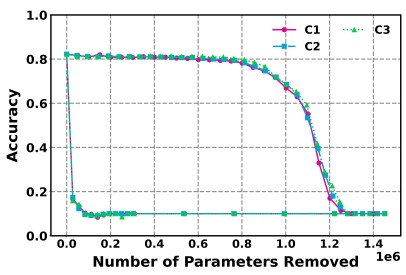

(a) Curvature-based pruning over three calibration sets.

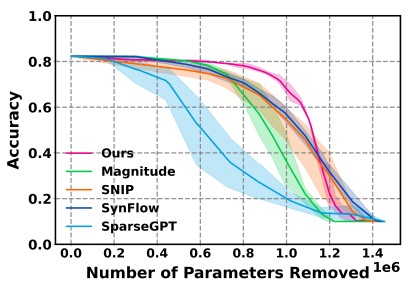

(b) Pruning results for all methods for three different CE models.

Figure 8: Ablation study (VGG9-lite, CE ReLU, CIFAR-10) with different random seeds.

Table 1: Runtime and memory comparison of pruning methods (VGG9-lite, CE ReLU, CIFAR-10). For all other methods, we record the statistics for one specific pruning sparsity. For ours, we record the average statistics over all ten examples.

| Method | Runtime (s) | Peak Memory (MB) | Reserved Memory (MB) |
|---|---|---|---|
| Magnitude | 0.010 | 32.1 | 130.0 |
| SNIP | 0.266 | 106.9 | 136.0 |
| SynFlow (100 iterations) | 20.580 | 59.1 | 140.0 |
| SparseGPT | 66.235 | 198.4 | 146.0 |
| Ours (1 example) | 1060.574 | 13493.0 | 368.0 |
| Magnitude pre-pruning + Ours (1 example) | 675.012 | 17201.2 | 368.0 |

to show our method is robust for different model states. Figure 8a shows the results using different calibration sets; as can be seen in the figure, our method is very robust to sampling variance in the calibration set. Figure 8b gives the results for different models, CIFAR-10 models (CE, ReLU), trained with different initialization states. We report the mean performance curve for each method and use the minimum and maximum values across three different models to define the error bounds. Although the curves vary with model changes, our method continues to outperform all other methods and achieves the smallest variability. All of the results remain consistent across runs, indicating that the proposed method is stable with respect to both the choice of calibration samples and model initialization.

# 7 Discussion and Conclusion

This paper proposed a differential-geometry-based approach to NN pruning, through identifying the main NN data flows. In particular, we introduced the notion of neural curvature that can be used to rank NN connections and separate them according to their importance. We provided an extensive evaluation over three image classification datasets.

**Computation requirements.** Since the proposed method requires additional computation as compared to existing approaches, we provide the runtime and GPU memory statistics for all considered methods in Table 1. For all of the other methods, since they need to specify the sparsity of the model pruning, we only record the runtime and GPU memory for one specific sparsity. For ours, we record the average runtime and GPU memory per example for VGG9-lite, CE ReLU, CIFAR-10.

**Scalability improvements.** In terms of scalability, the main challenge is calculating the Wasserstein distance, which requires solving a linear program. While this means that the Wasserstein distance cannot be computed in parallel on a GPU, we will explore three approaches for mitigating this challenge in future work. 1) We will consider a hybrid approach where magnitude-based pruning is used to quickly prune irrelevant weights (as assessed over a validation set) and curvature-based pruning is used for the challenging-to-separate, intermediate-value weights. An example of this procedure is shown in Figure 9. We begin by per-

forming magnitude pruning until the accuracy drops by more than a user-defined threshold, $\tau$, relative to the original accuracy; we set $\tau = 0.025$, which leads to $43.6\%$ of the model's parameters to be pre-pruned.

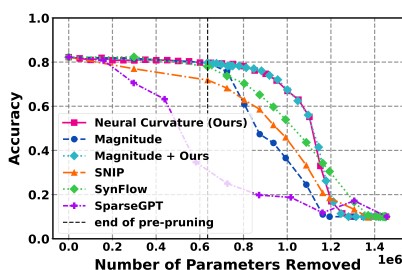

As illustrated in Figure 9, the black vertical dashed line indicates the switching point. Beyond this point, the performance of pure magnitude pruning degrades rapidly, whereas our method maintains accuracy and achieves performance comparable to the case without pre-pruning. We also show the runtime with pre-pruning in Table 1, which has a significant speedup. In future work, we will explore this approach in greater depth. 2) We will investigate approximations to the Wasserstein distance, which can be computed on the GPU, e.g., through Sinkhorn approximation (Luise et al., 2018). 3) We will improve the implementation by noting that the Wasserstein distance can be greatly simplified in a variety of cases, e.g., in input and output layers where it reduces to an inner product.

Figure 9: Scalability study (VGG9-lite, CE ReLU, CIFAR-10), adding Magnitude pre-pruning + Ours.

**Future work.** The symbolic nature of our approach provides a well-understood tool that can be used in a variety of applications such as robustness analysis, model repair and alignment. The method is general and can be used in a number of other domains, including natural language processing, reinforcement learning and magnetic resonance image analysis. As immediate future work, we intend to apply this technique to the adversarial training problem, e.g., by focusing on the edges that are currently not used by the NN. A similar approach can be used for model repair and alignment, e.g., by only re-training connections that are not currently used.

## Acknowledgments

This work was supported by the National Science Foundation (NSF) Grant CCF-2403615, the NSF Award 2243104 under the Center for Complex Particle Systems (COMPASS), the NSF Mid-Career Advancement Award BCS-2527046, the U.S. Army Research Office (ARO) under Grant No. W911NF-23-1-0111, the National Institute of Health (NIH) grants R01 AG 079957 and RF1 AG 082201, the Defense Advanced Research Projects Agency (DARPA) Young Faculty Award and DARPA Director Fellowship Award. Any opinions, findings, conclusions, or recommendations expressed in this material are those of the authors and do not necessarily reflect the views of the NSF or the US Government. This material is based upon work supported by the Air Force Office of Scientific Research under award number FA9550-25-1-0164.

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

# A Appendix

## A.1 Model Training

This section presents the model training details, including the training hyperparameters and model architectures.

## A.2 Training hyper-parameters

1. Training hyperparameters of CNN on MNIST and VGG9-lite on CIFAR-10:

   - learning rate: 2e-3 for all CE, WD, and AT models
   - weight decay parameter for WD models: 0.0001 (unless mentioned otherwise)
   - MNIST PGD parameter: eps = 0.1, $\alpha = 2/255$, iterations = 40
   - CIFAR-10 PGD parameter: eps = 2/255, $\alpha = 2/255$, iterations = 20
   - optimizer: Adam

2. Training hyperparameters of VGG9-lite on CIFAR-100

   - learning rate: 1e-3, 2e-3 for two CE models, 2e-3 for one WD model
   - weight decay parameter: 0.0001
   - optimizer: Adam

## A.3 CNN Architecture

The `CNN` model is a modified convolutional neural network inspired by the original LeNet-5 architecture. The network is designed for classification tasks and consists of two convolutional layers followed by three fully connected layers. The activation function (ReLU or Tanh) is applied after each convolution and linear transformation.

- **Input:** A grayscale image of size $28 \times 28$ (1 channel).

- **Layer 1: Convolution**

  - Convolutional layer with 6 output channels
  - Kernel size: $6 \times 6$
  - Stride: 2
  - Output size: $[6 \times 12 \times 12]$
  - Followed by ReLU/Tanh activation

- **Layer 2: Convolution**

  - Convolutional layer with 16 output channels
  - Kernel size: $6 \times 6$
  - Stride: 2
  - Output size: $[16 \times 4 \times 4]$
  - Followed by ReLU/Tanh activation

- **Flattening:** The $16 \times 4 \times 4$ output is flattened into a vector of size $256$.

- **Layer 3: Fully Connected**

  - Linear layer from 256 to 120 units
  - Followed by ReLU/Tanh activation

- **Layer 4: Fully Connected**

  - Linear layer from 120 to 84 units
  - Followed by ReLU/Tanh activation

- **Layer 5: Output Layer**
    - Linear layer from 84 to 10 units (representing 10 classes)

- **Output:** Raw logits for each class.

Note: The implementation uses custom `CNN` and `linear` methods for the forward computation, possibly to track graph-based properties such as edge connectivity and activation flows. Max-pooling operations are commented out in the current version.

## A.4 VGG9-lite Architecture

The `VGG9-lite` model is a customized version of the VGG architecture, adapted for the CIFAR-10/CIFAR-100 dataset. It contains 6 convolutional layers, followed by 3 fully connected layers. Batch normalization and ReLU/Tanh activations are used throughout. All the convolution and linear layers are computed using custom `CNN` and `linear` functions for graph-based analysis.

- **Input:** RGB image of size $32 \times 32$ (3 channels), normalized with CIFAR-10/CIFAR-100 dataset statistics.

- **Conv Block 1:**
    - One convolutional layers: $3 \rightarrow 64$ (kernel $3 \times 3$, padding 0, stride 2)
    - BatchNorm + ReLU/Tanh after each
    - Output size: $64 \times 16 \times 16$

- **Conv Block 2:**
    - One convolutional layers: $64 \rightarrow 128$ (kernel $3 \times 3$, padding 0, stride 1)
    - BatchNorm + ReLU/Tanh after each
    - Output size: $128 \times 14 \times 14$

- **Conv Block 3:**
    - Two convolutional layers: $128 \rightarrow 128$ (kernel $3 \times 3$, padding 0)
    - First convolutional layers with a stride 1; second convolutional layers with a stride 2
    - BatchNorm + ReLU/Tanh after each
    - Output size: $128 \times 5 \times 5$

- **Conv Block 4:**
    - Two convolutional layers: $128 \rightarrow 256$ (kernel $3 \times 3$, padding 0, stride 1)
    - BatchNorm + ReLU/Tanh after each
    - Output size: $256 \times 1 \times 1$

- **Flatten:** The $256 \times 1 \times 1$ tensor is flattened to a vector of length 256.

- **Fully Connected Layers:**
    - `fc1`: Linear layer from 256 to 512 (with ReLU/Tanh)
    - `fc2`: Linear layer from 512 to 128 (with ReLU/Tanh)
    - `fc3`: Linear layer from 128 to 10 (raw logits for 10 classes for CIFAR-10 (100 classes for CIFAR-100))

The use of custom `CNN` and `linear` functions enables explicit tracking of information flow and layer-wise graph structures, which is beneficial for graph-theoretic or curvature-based network analysis.

## A.5 Overall Algorithm

Algorithm 1 provides the pseudo-code of the overall algorithm of our method.

---

**Algorithm 1** Rank NN connections using neural curvature

---

**Input:** Trained neural network $f_\theta$, calibration set $\mathcal{D}$
**Output:** Curvature-ranked parameter set $edge\_set$
  1: $(V, E, w) \leftarrow \text{neural\_graph}(f_\theta)$                                   // Def. 6
  2: **Initialize** $edge\_cur[e] \leftarrow +\infty \quad \forall e \in E$
  3: **Initialize** $para\_set \leftarrow \emptyset$
  4: **for** $e \in E$ **do**
  5:     **for** $x \in \mathcal{D}$ **do**
  6:         Compute $\kappa_N(e, x)$                                              // Def. 9
  7:         $edge\_cur[e] \leftarrow \min\big(edge\_cur[e], \kappa_N(e, x)\big)$
  8:     **end for**
  9: **end for**
 10: // Fully-connected parameters correspond to single edges
 11: **for** fully-connected parameter $p \in \theta$ **do**
 12:     //Let $e_p \in E$ be the edge corresponding to $p$
 13:     $\kappa(p) \leftarrow edge\_cur[e_p]$
 14:     $para\_set[p] \leftarrow \kappa(p)$
 15: **end for**
 16: // For convolutional parameters, assign the minimum curvature among all incoming edges
 17: **for** convolutional parameter $p \in \theta$ **do**
 18:     $\mathcal{E}_p \leftarrow \{e \in E \mid e \text{ is induced by } p\}$
 19:     $\kappa(p) \leftarrow \min_{e \in \mathcal{E}_p} edge\_cur[e]$
 20:     $para\_set[p] \leftarrow \kappa(p)$
 21: **end for**
 22: // Rank parameters by curvature
 23: $edge\_set \leftarrow \text{sort}(para\_set, \text{descending})$                      // highest curvature first

---

## A.6 Proof of Proposition 1

**Proof 1** *To simplify notation, in what follows we define $u := v_{i,l}$ and $v := v_{j,l+1}$.*

Part 1: Edges in the input or output layers. *Without loss of generality, assume $u$ is in the input layer. This means the probability mass assigned to $u$ equals $1$. Consider the set of nodes involved in the Wasserstein distance computation (ordered for simplicity): $\{u, v, \Gamma^{out}(v)\}$. In this case, $\mu_u^\alpha(\cdot, x) = [1, 0, \ldots, 0]$ and $\mu_v^\alpha(\cdot, x) = [0, \alpha, e_{v_1}, \ldots, e_{v_N}]$, where $\sum e_{v_i} = 1 - \alpha$. Thus, the Wasserstein distance between $\mu_u^\alpha(\cdot, x)$ and $\mu_v^\alpha(\cdot, x)$ is equal to the cost of transferring a mass of $\alpha$ from $u$ to $v$ and a mass of $(1 - \alpha)$ from $u$ to $\Gamma^{out}(v)$, i.e., the neighbors of $v$ in layer 2. When $d_\sigma$ is sufficiently large, all shortest paths from $u$ to $\Gamma^{out}(v)$ are cheaper than $d_\sigma$, hence:*

$$W(\mu_u^\alpha(\cdot, x), \mu_v^\alpha(\cdot, x)) = d_\sigma(u, v, x)\,\alpha + r\,(1 - \alpha),$$

*where $r = W_{-(u,v)}(\mu_u^1(\cdot, x), \mu_v^0(\cdot, x))$, i.e., the transport cost from $u$ to $\Gamma^{out}(v)$, without using the edge $(u, v)$. Substituting into the definition of neural curvature gives*

$$
\begin{aligned}
\kappa_N(u, v, x) &= \lim_{\alpha \to 1} \frac{1 - \frac{W(\mu_u^\alpha(\cdot,x), \mu_v^\alpha(\cdot,x))}{d_\sigma(u,v,x)}}{1 - \alpha} \\
&= \lim_{\alpha \to 1} \frac{1 - \alpha - \frac{r(1-\alpha)}{d_\sigma(u,v,x)}}{1 - \alpha} \\
&= \lim_{\alpha \to 1} \left( 1 - \frac{r(1 - \alpha)}{d_\sigma(u, v, x)(1 - \alpha)} \right) \\
&= 1 - \frac{r}{d_\sigma(u, v, x)}.
\end{aligned}
$$

*Taking the limit yields*

$$\lim_{d_\sigma(u,v,x) \to \infty} \kappa_N(u, v, x) = 1.$$

Part 2: Edges in hidden layers. *Consider the set of nodes involved in the Wasserstein distance computation (ordered for simplicity): $\{\Gamma^{in}(u), u, v, \Gamma^{out}(v)\}$. In this case, $\mu_u^\alpha(\cdot, x) = [e_{u_1}, \ldots, e_{u_M}, \alpha, 0, \ldots, 0]$ and $\mu_v^\alpha(\cdot, x) = [0, \ldots, 0, \alpha, e_{v_1}, \ldots, e_{v_N}]$, where $\sum e_{u_i} = 1 - \alpha$ and $\sum e_{v_i} = 1 - \alpha$. In this case, as $d_\sigma$ gets large, the Wasserstein distance simplifies to three terms: i) the cost of transporting a mass of $(1 - \alpha)$ from $\Gamma^{in}(u)$ to $v$, without using edge $(u, v)$; ii) the cost of transporting a mass of $\alpha - (1 - \alpha)$ from $u$ to $v$; (iii) and the cost of transporting a mass of $(1 - \alpha)$ from $u$ to $\Gamma^{out}(v)$, without using edge $(u, v)$. Thus, the Wasserstein distance can be expressed as*

$$W(\mu_u^\alpha(\cdot, x), \mu_v^\alpha(\cdot, x)) = r_1(1 - \alpha) + d_\sigma(u, v, x)\big(\alpha - (1 - \alpha)\big) + r_2(1 - \alpha),$$

*where $r_1 = W_{-(u,v)}(\mu_u^0(\cdot, x), \mu_v^1(\cdot, x))$ and $r_2 = W_{-(u,v)}(\mu_u^1(\cdot, x), \mu_v^0(\cdot, x))$, with $W_{-(u,v)}$ interpreted as in Part 1. Substituting into the neural curvature definition yields*

$$
\begin{aligned}
\kappa_N(u, v, x) &= \lim_{\alpha \to 1} \frac{1 - \frac{W(\mu_u^\alpha(\cdot,x), \mu_v^\alpha(\cdot,x))}{d_\sigma(u,v,x)}}{1 - \alpha} \\
&= \lim_{\alpha \to 1} \frac{1 + 1 - 2\alpha - \frac{(r_1 + r_2)(1 - \alpha)}{d_\sigma(u,v,x))}}{1 - \alpha} \\
&= \lim_{\alpha \to 1} \left( \frac{2(1 - \alpha)}{1 - \alpha} - \frac{r_1 + r_2}{d_\sigma(u, v, x))} \right) \\
&= 2 - \frac{r_1 + r_2}{d_\sigma(u, v, x)}.
\end{aligned}
$$

*Taking the limit gives*

$$\lim_{d_\sigma(u,v,x) \to \infty} \kappa_N(u, v, x) = 2.$$

## A.7 Pruning-Based Comparison (Tanh)

We present the results for Tanh models in this section on the MNIST and CIFAR-10 datasets. We remove the SynFlow curve in Figure 10 since it is designed for Tanh activations.

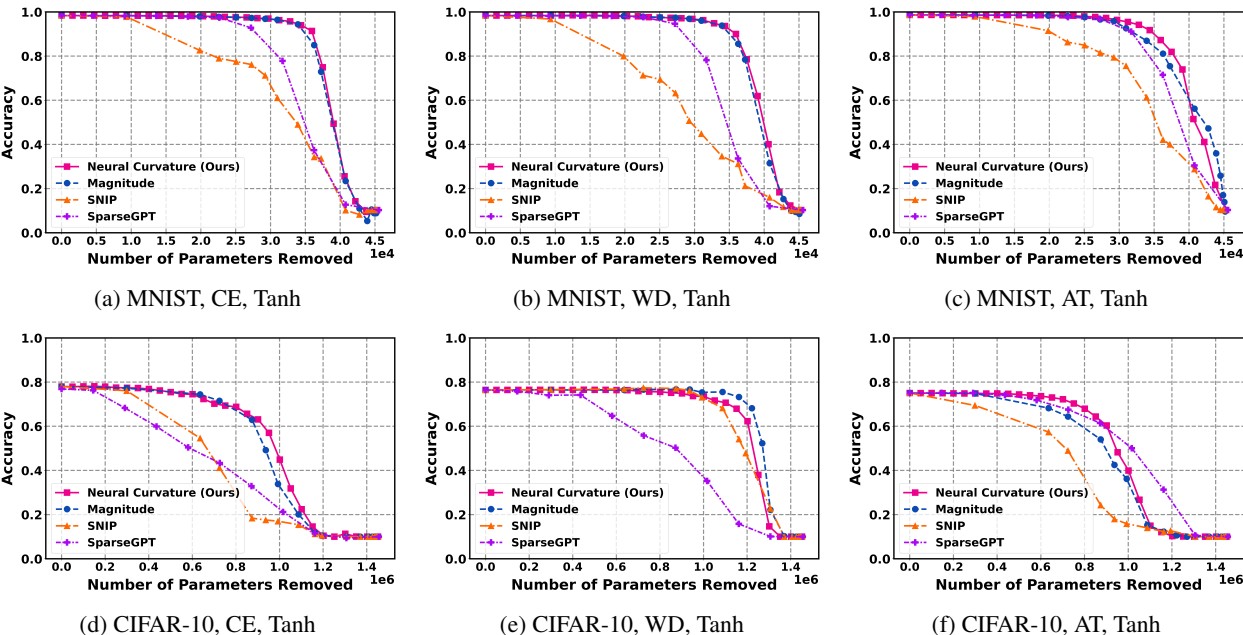

Figure 10: Edge removal evaluation on MNIST, CIFAR-10, and CIFAR-100. Each subfigure shows a comparison of our neural curvature algorithm with magnitude and SNIP pruning methods.

## A.8 Edge Removal Evaluation

In this section, we present the results for all the models, including CNN on MNIST, VGG9-lite on CIFAR-10 and CIFAR-100 (Figure 11. Each subfigure contains the full model edge removal results based on neural curvature values using positive-first and negative-first edge sets. The number on the curve shows the curvature value at that data point.

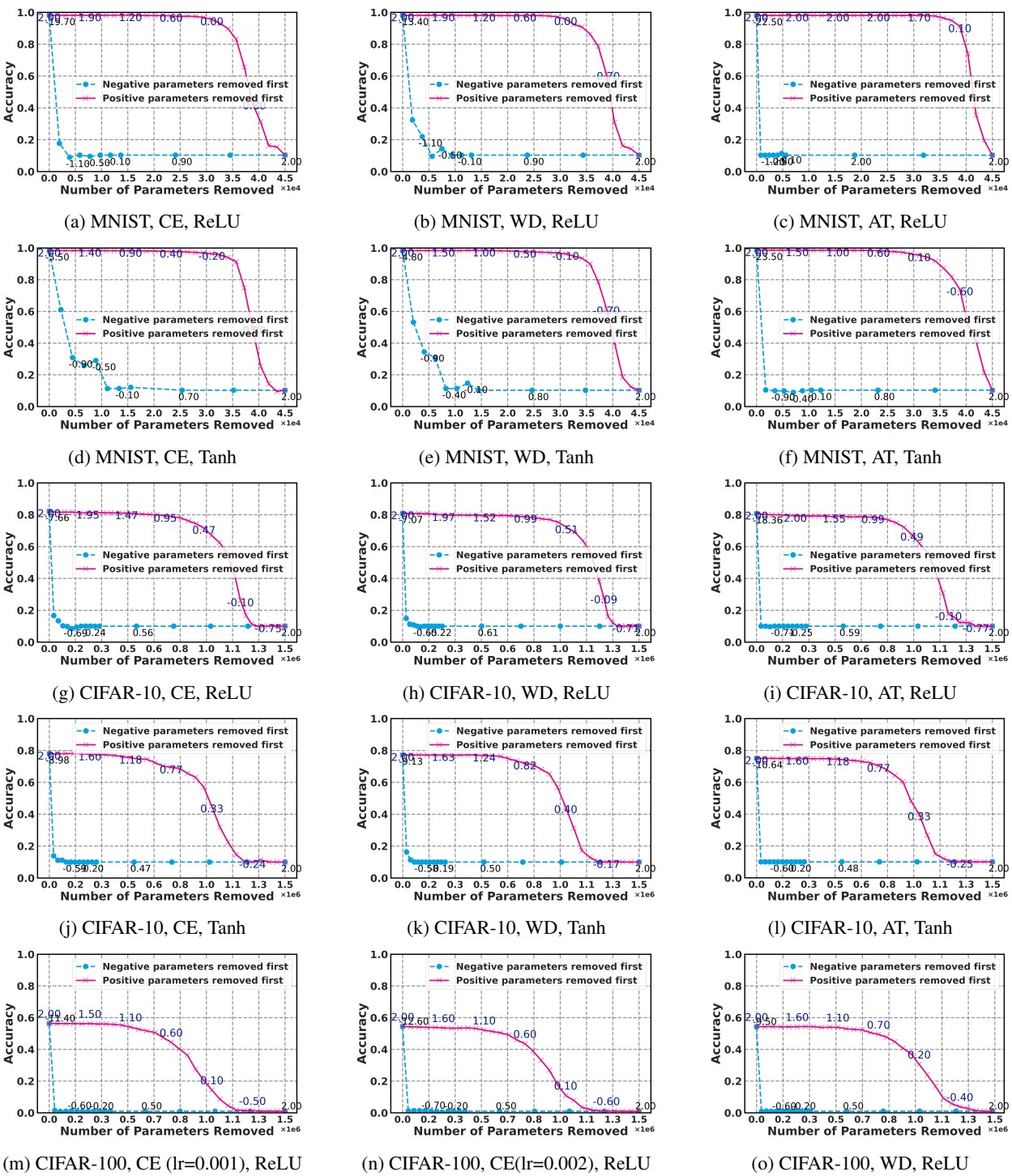

Figure 11: Edge removal evaluation on MNIST, CIFAR-10, and CIFAR-100.

## A.9   Ablation Study of WD hyper-parameter of Tanh models

Figure 12 presents results for VGG9-lite Tanh, trained with WD parameters $1e-5$, $1e-4$, and a mixed $1e-4/1e-5$ setup to achieve higher accuracy.

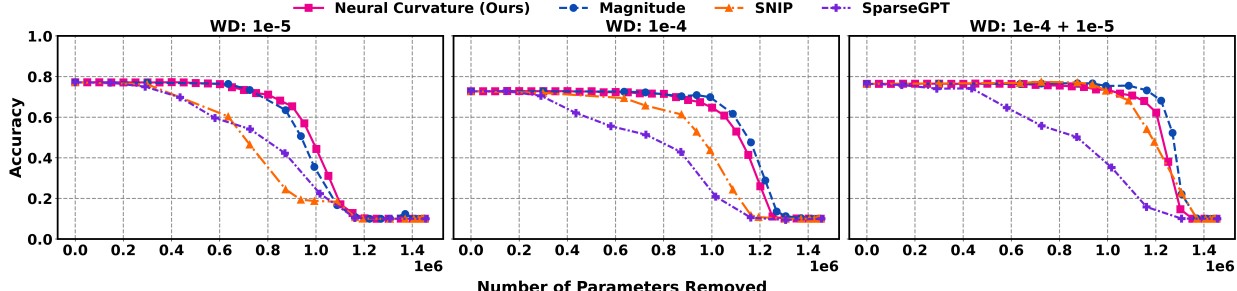

Figure 12: Full model edge removal comparison between magnitude-based and ours (VGG9-lite, WD Tanh, on CIFAR-10). Models are trained with different WD parameters.

## A.10 Ablation Study of the iteration number of SynFlow

Figure 13 presents comparison results between SynFlow and ours, for VGG9-lite ReLU models, on CIFAR-10 and CIFAR-100, where we explore the number of iterations of SynFlow, using 100, 50, and 5, to get a better evaluation. As shown in Figure 13, the performance of SynFLow with 100 and 50 iterations is almost the same. However, when we reduce the number of iteration to 5, the performance of SynFlow improves, and this is consistent in all the models, which suggests that the recommended setting in the original paper by Tanaka et al. (2020) may not be optimal for all models.

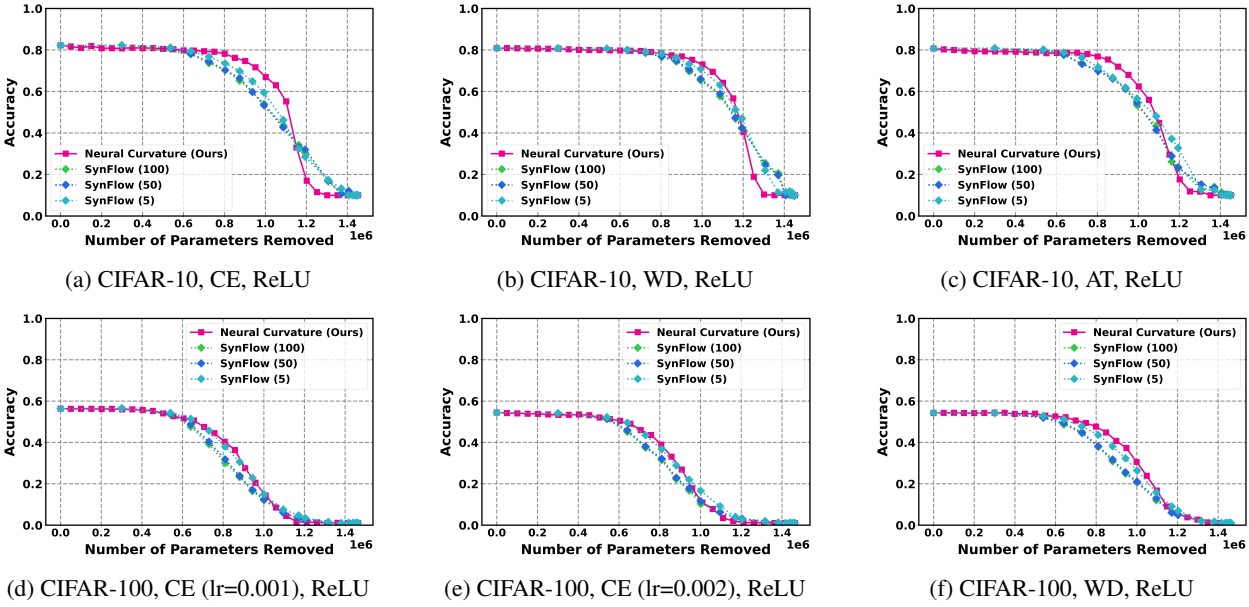

(a) CIFAR-10, CE, ReLU      (b) CIFAR-10, WD, ReLU      (c) CIFAR-10, AT, ReLU

(d) CIFAR-100, CE (lr=0.001), ReLU      (e) CIFAR-100, CE (lr=0.002), ReLU      (f) CIFAR-100, WD, ReLU

Figure 13: Edge removal evaluation CIFAR-10, and CIFAR-100 (lr means learning rate). Each subfigure shows a comparison of our neural curvature algorithm with SynFlow pruning method, which was applied with 100, 50, and 5 iterations.

### A.11    Ablation Study of Per-layer edge removal

In this section, we presented the results of the ablation study on per-layer edge removal, including CNN on MNIST (Figure 14, and Figure 15), VGG9-lite on CIFAR-10 (Figure 16, Figure 17). Each figure contains the per-layer edge removal results based on neural curvature values (top) and based on weight magnitude (bottom), for comparison.

An interesting observation is that when pruning is restricted to a single layer at a time, our method behaves similarly to magnitude pruning. This suggests that within a given layer, the largest-magnitude weights are typically the most important. However, magnitude alone cannot be meaningfully compared across layers, which explains why our method demonstrates significantly stronger performance when pruning the full network: the neural curvature provides an effective normalization across layers and enables globally coherent pruning decisions.

Moreover, the curvature annotations along our pruning curves reveal that model accuracy typically begins to degrade only once the edges being removed have curvature values near or below zero. This layer-wise pattern again validates curvature as a meaningful importance indicator. Finally, this experiment highlights that our approach is capable of reliably identifying both the most and least important edges across the network.

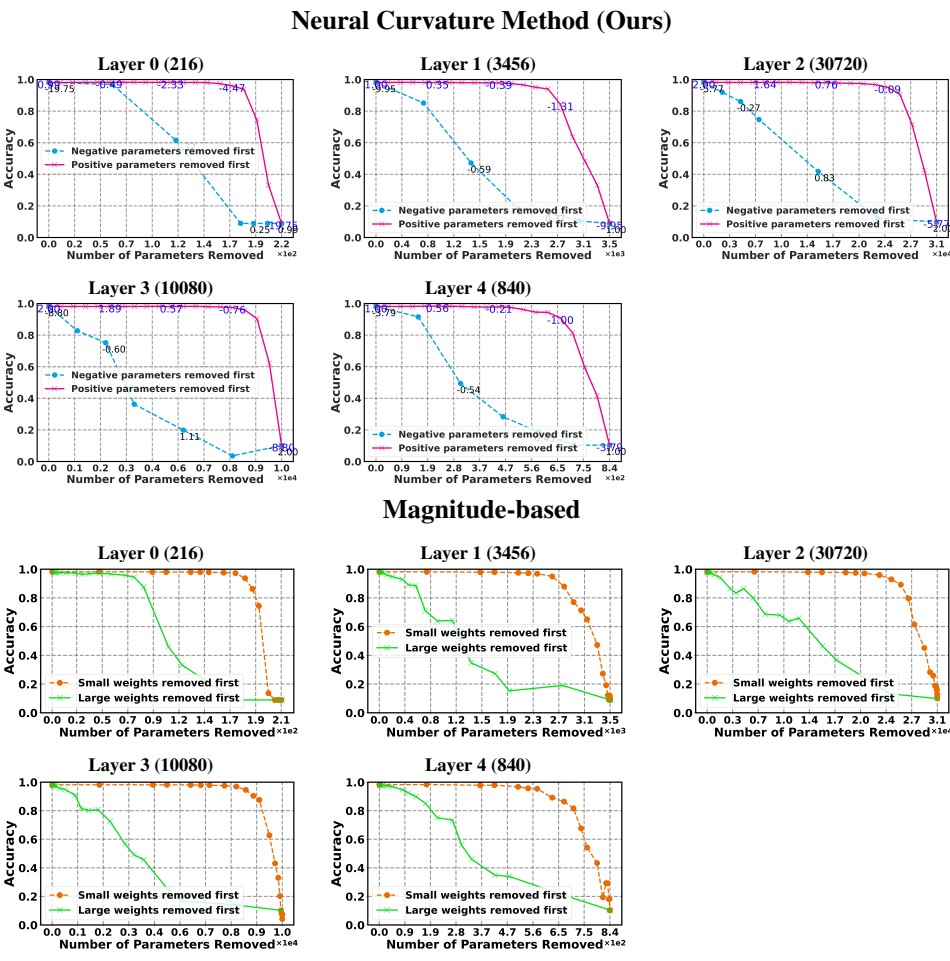

Figure 14: Ablation experiment on per-layer edge removal. Each subfigure shows the edge removal analysis for each layer based on neural curvature value (top 5 figures) and magnitude-based pruning (bottom 5 figures) for the CNN, CE ReLU configuration. The number above the figure is the total number of parameters in that layer.

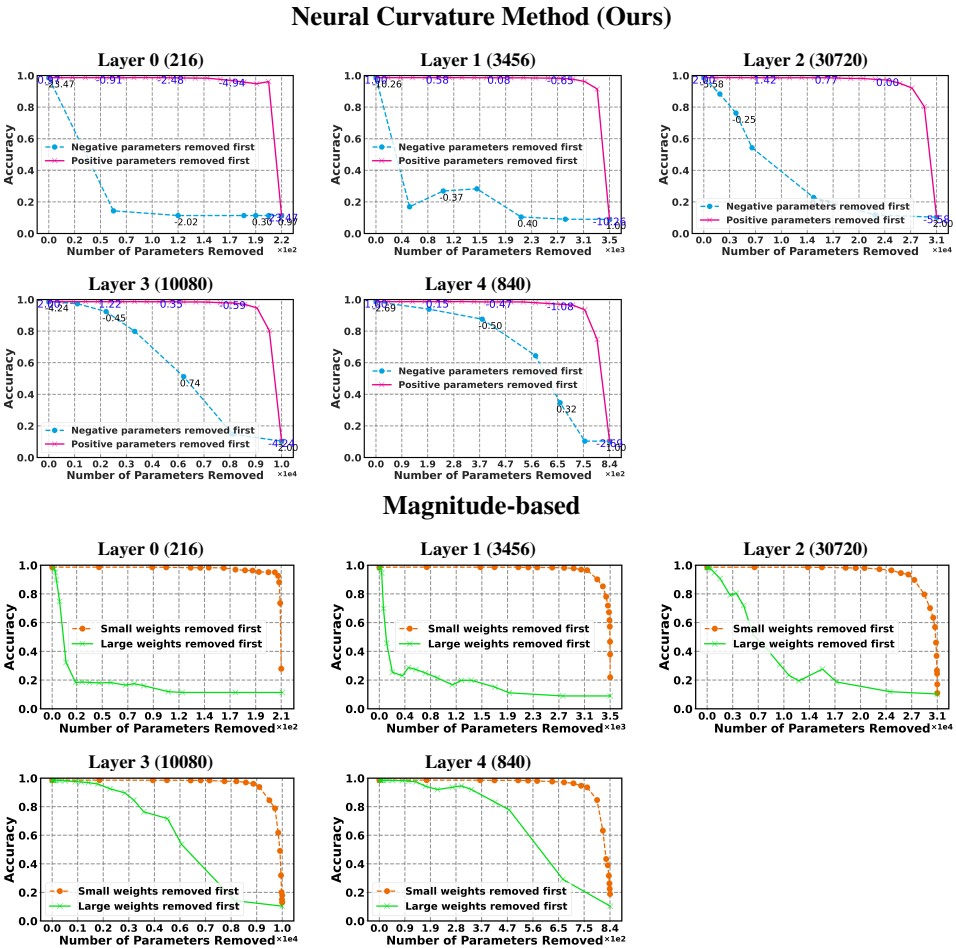

Figure 15: Ablation experiment on per-layer edge removal. Each subfigure shows the edge removal analysis for each layer based on neural curvature value (top 5 figures) and magnitude-based pruning (bottom 5 figures) for the CNN, AT Tanh configuration. The number above the figure is the total number of parameters in that layer.

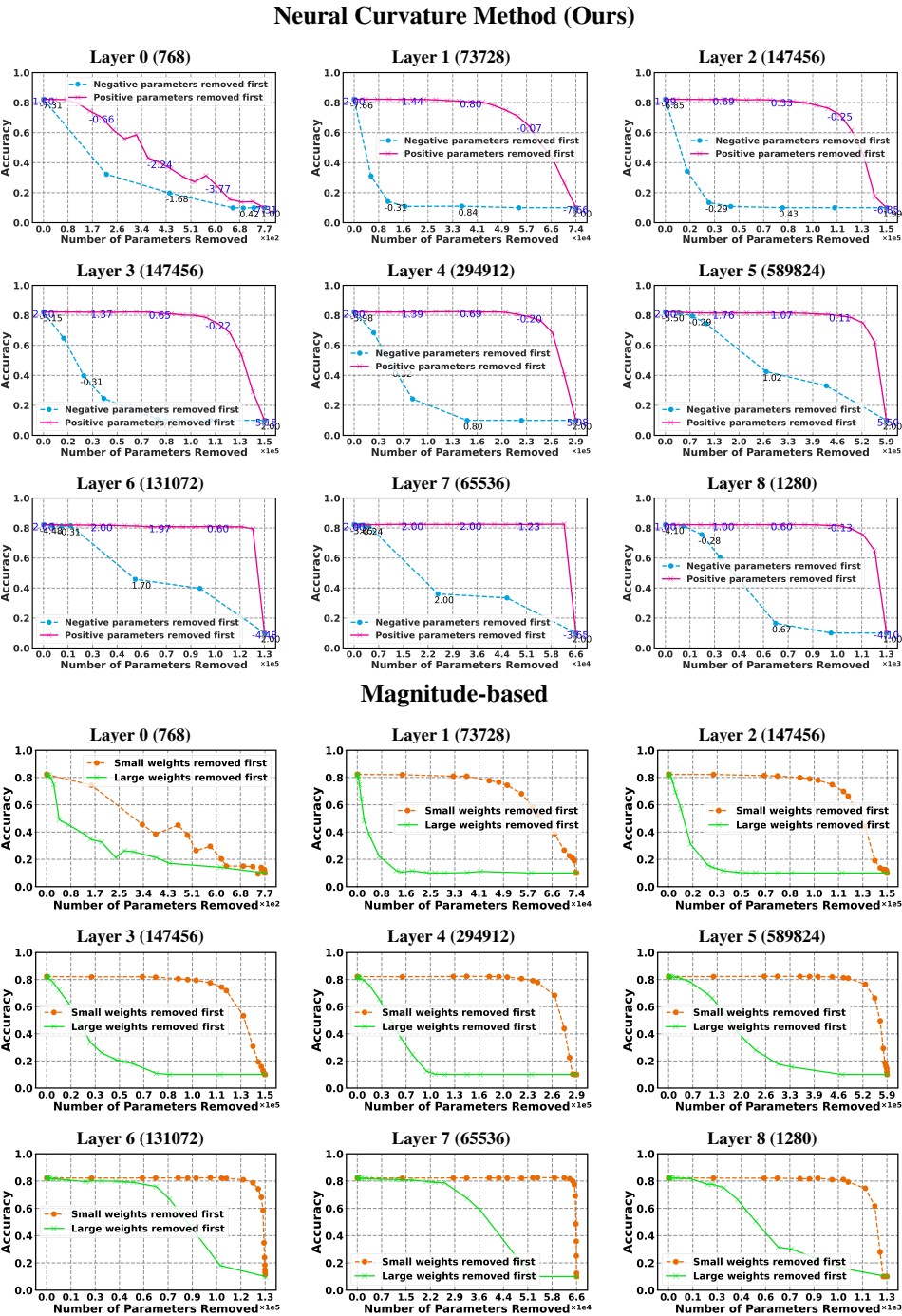

Figure 16: Ablation experiment on per-layer edge removal. Each subfigure shows the edge removal analysis for each layer based on neural curvature value (top 9 figures) and magnitude-based pruning (bottom 9 figures) for the VGG9-lite, CE ReLU configuration. The number above the figure is the total number of parameters in that layer.

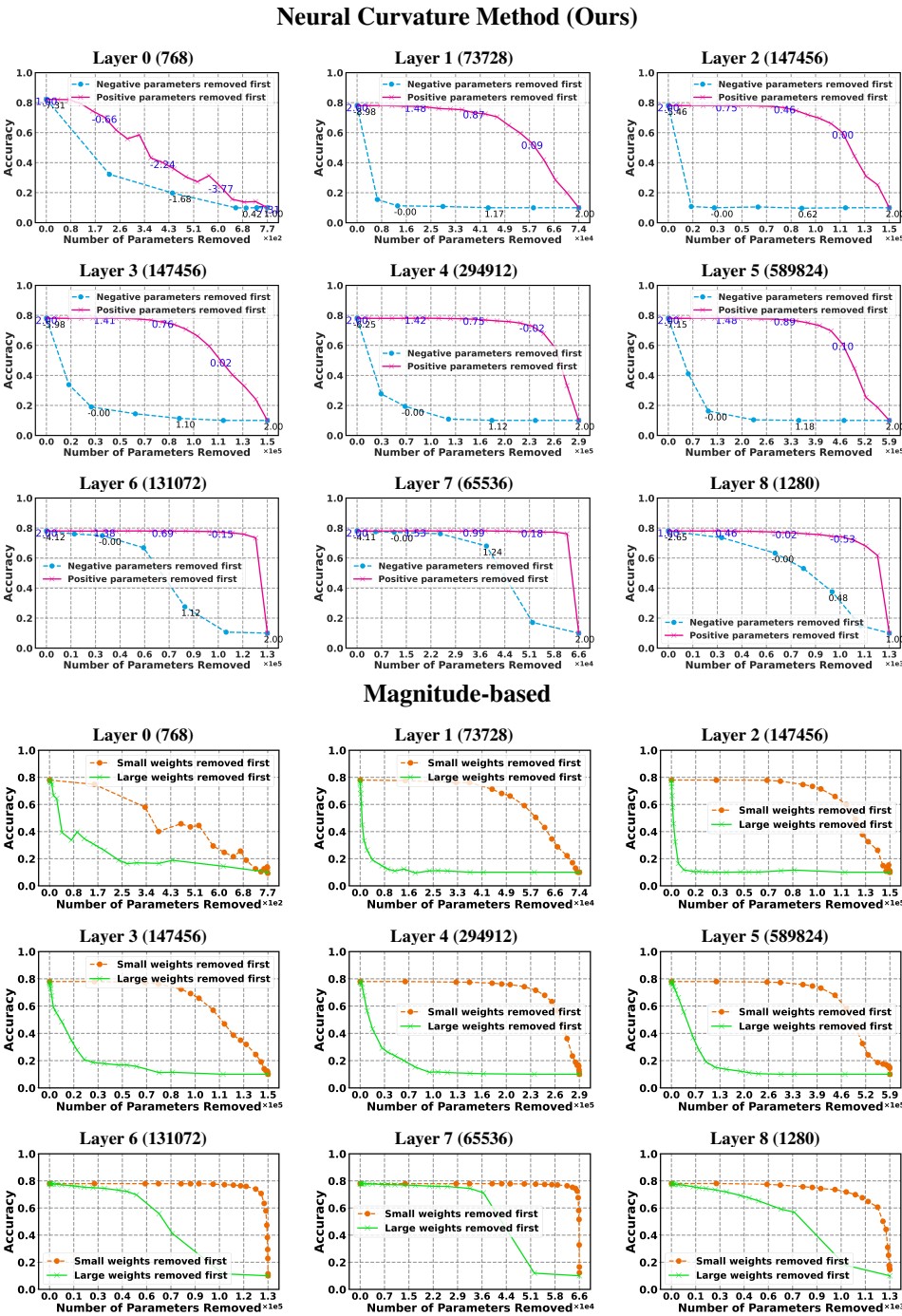

Figure 17: Ablation experiment on per-layer edge removal. Each subfigure shows the edge removal analysis for each layer based on neural curvature value (top 9 figures) and magnitude-based pruning (bottom 9 figures) for the VGG9-lite, CE Tanh configuration. The number above the figure is the total number of parameters in that layer.

