# OpenReview forum: "Post-Training Neural Network Pruning using Graph Curvature"
_TMLR — Accepted by TMLR_

### Review · Reviewer_vpzj · 2026-01-27

**Summary Of Contributions:**

## Summary

This paper proposes a method to measure the strength of information flow exchanged among neural network parameters by modifying and adapting the concept of Ollivier–Ricci curvature (ORC) from differential geometry. ORC assigns positive values to nodes that are highly connected to many other nodes, and negative values to nodes that are sparsely connected but serve as essential bottlenecks through which traffic between other nodes must pass. The paper hypothesizes that neurons with negative ORC values are more important, while those with positive values are less important. Based on this hypothesis, the authors estimate neuron importance and apply it to pruning, achieving better performance than the baseline methods considered.

## Strengths
- The problem is approached in an interesting way. Compared to existing heuristic-based methods that rely only on magnitude or gradient values, this approach appears to have stronger theoretical grounding.
- Pruning is an important and timely problem given the increasing size of modern models.

## Weaknessese
### Presentation
- Although the title is “Analyzing Neural Network Information Flow,” the paper does not actually perform much analysis. Overall, it appears closer to a paper proposing a new pruning method based on differential geometry. If the title is to be justified, the paper would need to theoretically define information flow, derive its properties through theorems, and empirically verify whether those properties hold in practice. Such attempts are not clearly present.
- There are many parts that could be removed or moved to the appendix, as well as substantial redundancy, which makes the paper unnecessarily long. For example, Algorithm 1 and the proof in Proposition 1 would be more appropriate for the appendix, and Remark 3 and Subsection 5.3 appear to cover nearly identical content. Similarly, Figure 3 would benefit from presenting only the main results in the main text, with the rest moved to the appendix. The same applies to Figure 5. (While there may not be strict page limits, the current format hurts readability.)
- Sections 3 and 6 seem too short to justify having their own sections.

### Method
- Overall, the method design lacks explanations addressing the “why” behind key choices. This disrupts the logical flow of the research design. For example:
  - The formula in Definition 6 is highly unintuitive. The final norm function appears to resemble some form of cross-entropy; it is unclear why neuron values cannot simply be inserted directly.
  - It is unclear why the formulation in Definition 7 is derived in that particular way.
  - When computing values by plugging in samples $x$, it is unclear why the minimum is used rather than the mean or maximum.

### Experiments
- Many of the baseline methods are either outdated or evaluated under unfair settings. For instance, as the authors themselves note, SNIP and SynFlow are methods intended for use at initialization, yet they are applied to trained models here. This is problematic because small gradients before training may indicate undertraining, whereas small gradients after training may indicate successful convergence. It is unclear whether these methods should be described as state-of-the-art.
- The curvature is currently computed by sampling from the validation set, but it seems more appropriate to compute it on the training set.
- It is unclear how the calibration set is sampled. Relatedly, it would be interesting to see how different sampling for the calibration set affect results; error bars could be added to the figures.
- It is unclear whether CIFAR-100 experiments were conducted only under limited settings compared to other datasets.
- The rationale behind what Section 7.3.2 is intended to test is unclear.

### Minor Points / Questions
- In Example 1, the formula for computing the distribution $m$ is difficult to understand.
- In the proof of Proposition 1, it is unclear why the terms corresponding to going from $u$ to $v$, from $u$ to $\Gamma^{out}$, and from $\Gamma^{in}$ to $v$ are computed as independent terms. Shouln'td the constant $r$ be expressed as a function of $d$?
- In Figure 3, it would be better to remove SynFlow, which is not implementable in this setting. Although the method is not actually applied, it is depicted as performing poorly.

## Typos
- Section 6, third paragraph: “layer $t$” → “layer $k$”

**Audience:**

Yes

**Audience Explanation:**

Pruning is an important problem, especially recently.

**Claims And Evidence:**

No

**Claims Explanation:**

Please check the weakness part of the previous answer.

**Requested Changes:**

Please improve the readability, logical flow, and experimental rigor as suggested.

---

> ### Author Response · Authors · 2026-03-03
> **Response**
>
> ## Presentation
> ### Although the title is “Analyzing Neural Network Information Flow,” the paper does not actually perform much analysis.
>
> **1. Thank you for this comment. Following the reviewer's suggestion, in the revised version, we have improved the paper's scope and are now focusing exclusively on post-training pruning using graph curvature. This is reflected both in the paper's new title and updated introduction, as noted in the main response.**
>
> ### There are many parts that could be removed or moved to the appendix
>
> **2. Thank you for this helpful comment. We agree with the reviewer and have revised the manuscript accordingly. We moved the proof of Proposition 1 and Algorithm 1 to the Appendix and removed the redundant Remark 3. We also modified Figure 3: we only kept the results for all the ReLU models in Figure 3, and moved the Tanh model results to the appendix. Furthermore, we moved Subsection 7.3.2 and Figure 5 to the appendix.**
>
> ### Sections 3 and 6 seem too short to justify having their own sections.
>
> **3. Thank you for this suggestion. We merged Section 6 into Section 5. We prefer to keep Section 3 as a separate section as it makes explicit the specific problem addressed in the paper.**
>
> ## Method
> ### The formula in Definition 6 is highly unintuitive.
> **1. Thank you for pointing out this. We adjusted Definition 6 (Definition 7 in the revised version) to make it easier to understand. We added Definition 2, the definition of exponential normalization of neighbor distributions, which serves as the original graph-based intuition behind Definition 7. Indeed, the neuron values are used in the normalization function in Definition 7 -- this is made explicit in the revised version.
> The normalization aims to deal with the potential numeric issue caused by ReLU, as used in an exponential function. It can help smooth the node value distribution, get a better distinguishing difference among vertices while ensuring a valid probability measure on each neighborhood, which brings better results.}% Otherwise, some nodes with large values would dominate the probability measure space.**
>
> ### It is unclear why the formulation in Definition 7 is derived in that particular way.
>
>    **2. Thank you for your valuable comment. We added an explanation for Definition 7 (now Definition 8) in the paper. Definition 8 is introduced to capture the nonlinearity of the activation function. When the activation of the target node is zero for ReLU, or saturated and close to one for Tanh, the transmission of information (i.e., the edge weight) is effectively blocked. Definition 8 is designed to reflect this phenomenon.**
>
>   **The nonlinearity is encoded by $\beta(x)$. For ReLU, $\beta(x)=0$ when a node is inactive, so no information can pass through that node; this applies to both the source and the target. For Tanh, when $\beta(x)$ is close to $0$, the target node is saturated and the transmitted information is clipped. In both cases, the effective edge cost increases, which motivates the rescaling $\frac{w_{f_\theta}}{\beta}$.**
>
>  **For ReLU, the edge becomes ineffective if either the source or the target node is inactive; therefore, we use $\min(\beta(x_{source}), \beta(x_{target}))$.
>     For Tanh, only saturation at the target node limits the transmission, so we use $\beta(x_{target})$. We have clarified this relationship in the text before Definition 8 in the paper.**
>
> ### When computing values by plugging in samples $x$, it is unclear why the minimum is used rather than the mean or maximum.
>    **3. Thank you for your helpful comment. We use the minimum curvature over the samples because more negative curvature indicates a stronger contribution to information flow and model performance. Taking the minimum, therefore, captures the most critical role that a connection can play for any input, rather than its average behavior. In contrast, the mean would dilute these highly informative cases, especially when the edge is only activated for a subset of samples, and the maximum would emphasize the least informative regime.**
>
>   **From a pruning perspective, we aim to identify connections that are essential for at least some data points. If an edge exhibits highly negative curvature for certain inputs, removing it may significantly harm performance even if its average curvature is moderate. The minimum thus provides a conservative and importance-aware estimate of the edge’s functional contribution across the dataset. We have clarified these points in Section 5.4.**

---

> ### Author Response · Authors · 2026-03-03
> **Response**
>
> ## Experiments
> ### Many of the baseline methods are either outdated or evaluated under unfair settings.
> **1. Thank you for pointing this out. We do agree that comparing against methods designed for a prune-and-retrain iteration may be incomplete. For better evaluation, we have also included a comparison with SparseGPT, a state-of-the-art post-training pruning method, as described in the main response. Our method outperforms SparseGPT on all considered models as well.**
>
> **We also note that magnitude pruning is in fact a very effective post-training pruning method given the right training hyper-parameters. As noted in a paper on this topic [1], magnitude pruning can achieve comparable performance to most state-of-the-art methods as long as the training hyper-parameters are chosen accordingly. Since our approach outperforms magnitude pruning in a variety of training settings, we believe this is a strong indication that our method is indeed on par with state-of-the-art methods.**
>
> - [1] Wang H, Qin C, Bai Y, Fu Y. Why is the state of neural network pruning so confusing? On the fairness, comparison setup, and trainability in network pruning. arXiv preprint arXiv:2301.05219. 2023 Jan 12.
>
>
> ### The curvature is currently computed by sampling from the validation set, but it seems more appropriate to compute it on the training set.
>  **2. Thank you for your valuable comment. We agree that calculating curvatures using the training set would be more data efficient. Since the model is trained on the training set, the feature propagation patterns of these samples are already fitted by the network. Computing curvature on the training data would therefore reflect information flow that the model has already memorized, potentially leading to an overly optimistic and biased estimate.**
>
> **In contrast, the validation set is unseen during training. The fact that the curvature computed from these samples still captures meaningful information flow demonstrates that the proposed measure is not data-specific and can generalize beyond the training distribution. This is particularly important for pruning in deployment, where the model must operate on unseen data. Using validation samples, therefore, provides a more faithful assessment of robustness and improves the practical generalization of the method.**
>
>
> ### It is unclear how the calibration set is sampled. Relatedly, it would be interesting to see how different sampling for the calibration set affect results; error bars could be added to the figures.
> **3. Thank you for your helpful suggestion. The calibration set corresponds to the validation set split from the training data. The samples used for curvature computation are randomly drawn from this validation set and aggregated by label.**
>
> **To evaluate the robustness of the proposed method with respect to the sampling procedure, we have added new results obtained using different random seeds for selecting the calibration samples. The corresponding statistics are reported in the ablation study. The results show consistent performance across different seeds, indicating that the method is stable with respect to the choice of calibration samples.**
>
> ### It is unclear whether CIFAR-100 experiments were conducted only under limited settings compared to other datasets.
> **4. Thank you for pointing this out. To provide a more comprehensive evaluation on CIFAR-100, we trained three additional models: two using cross-entropy loss with different learning rates, and one using weight decay with the same hyperparameter settings as the other datasets. The corresponding results have been added to Figure~3. These additional experiments show consistent trends with the other datasets, confirming that the conclusions are not specific to a limited setting.**
>
> ### The rationale behind what Section 7.3.2 is intended to test is unclear.
> **5. Thank you for the helpful comment. We have moved Section~7.3.2 to the appendix for better organization. The goal of this experiment is to evaluate whether the proposed method can more effectively identify important and unimportant parameters across layers.**
>
> **Magnitude-based pruning can achieve comparable or slightly better results when the pruning ratio is fixed independently for each layer. However, when pruning is performed globally over the entire model, our method significantly outperforms magnitude pruning. This indicates that our approach provides a more reliable cross-layer importance measure. This demonstrates that the proposed curvature-based criterion yields a more consistent global ranking of parameter importance.**

---

> ### Author Response · Authors · 2026-03-03
> **Response**
>
> ## Minor Points / Questions
> ### In Example 1, the formula for computing the distribution $m$ is difficult to understand.
> **1. Thank you for pointing this out. In Example~1, all probability measures $m$ have the same length, as they are defined over the full set of nodes in the graph (11 in this case). For instance, node $n_5$ has five neighbors, each connected with unit weight, and therefore each neighbor is assigned the same probability. Since the probabilities sum to one, each neighbor has mass $1/5 = 0.2$, while all non-neighboring nodes have probability zero. We also added an explanation for this example in the paper.**
>
> ### In the proof of Proposition 1, it is unclear why the terms corresponding to going from $u$ to $v$, from $u$ to$\Gamma^{out}$, and from $\Gamma^{in}$ to $v$ are computed as independent terms. Shouldn't the constant $r$be expressed as a function of $r$?
> **2. Thank you for pointing this out. To clarify the proof, we have explicitly defined the distributions involved in the Wasserstein computation. For example, in the case where $u$ is in the input layer, the set of nodes involved in the Wasserstein distance computation is (ordered for simplicity): $\{u, v, \Gamma^{out}(v)\}$. In this case, $\mu_u^{\alpha}(\cdot, x) = [1, 0, \dots, 0]$ and $\mu_v^{\alpha}(\cdot, x) = [0, \alpha, e_{v_1}, \dots, e_{v_N}]$, where $\sum e_{v_i} = 1-\alpha$. Thus, the Wasserstein distance between $\mu_u^{\alpha}$ and $\mu_v^{\alpha}$ is equal to the cost of transferring a mass of $\alpha$ from $u$ to $v$ and a mass of $(1-\alpha)$ from $u$ to $\Gamma^{out}(v)$, i.e., the neighbors of $v$ in layer 2.**
>
> ### In Figure 3, it would be better to remove SynFlow, which is not implementable in this setting. Although the method is not actually applied, it is depicted as performing poorly.
> **3. Thank you for the suggestion. We agree with that and removed the  Syflow curve for the Tanh models. Currently, the results for Tanh models are moved to the Appendix.**
>
>
> ## Typos
> **Thank you for catching this typo. We have corrected “layer $t$” to “layer $k$” in Section 6 (now it is Section 5.5).**

---

> > ### Comment · Reviewer_vpzj · 2026-03-03
> >
> > Thank you for the responses. It helped me a lot to better understand the manuscript.
> >
> > Some quick questions:
> >
> > - What is $w(u, v)$? it is used in Def. 1 (as an example of cost) and Def. 2 (to define $V_x$), but not sure what it is. As a result, it is difficult to capture what 'value function' is (appearing in Def. 2 and Def. 7.)
> >
> > - Also, two notations of $V$, vertex set $V_l$ and value function $V_x$ and $V_n$, are confusing. Need clarification.
> >
> > > When computing values by plugging in samples $x$, it is unclear why the minimum is used rather than the mean or maximum.
> >
> > - Regarding this point, I guess an additional ablation study would be beneficial. Additionally, regarding the multiple random seed and calibration set ablation study, it would be better to visualize with error bars or coloring standard deviation span, rather than displaying separate plots for each seeds... I understand there are many baseline models, so more consideration would be required.
> >
> > - It is interesting that SparseGPT performs so poorly compared to magnitude based methods, unlike original paper. Do you have any guess?

---

> > > ### Author Response · Authors · 2026-03-05
> > > **Response**
> > >
> > > - What is $w(u,v)$? it is used in Def. 1 (as an example of cost) and Def. 2 (to define
> > > $V_x$), but not sure what it is. As a result, it is difficult to capture what 'value function' is (appearing in Def. 2 and Def. 7.)
> > >
> > > **Thanks for your comment. The function $w: E \to \mathbb{R}_+$ defines each edge weight in the graph, as introduced in Section 3.**
> > >
> > > - Also, two notations of $V$, vertex set $V_t$ and value function $V_x$ and $V_n$, are confusing. Need clarification.
> > >
> > > **Thanks for your comments. Indeed, there was confusion regarding notation. In the revised version, $V$ remains the vertex set of the graph (with $V_l$ the vertex set for layer $l$). The node value function, previously $V_n$ and $V_n$, is now changed to $h_n$ and $h_x$ to avoid confusion.**
> > >
> > > ### When computing values by plugging in samples, it is unclear why the minimum is used rather than the mean or maximum.
> > >
> > >    - Regarding this point, I guess an additional ablation study would be beneficial. Additionally, regarding the multiple random seed and calibration set ablation study, it would be better to visualize with error bars or coloring standard deviation span, rather than displaying separate plots for each seeds... I understand there are many baseline models, so more consideration would be required.
> > >
> > > **Thanks for your suggestion. We present the mean and error bound for each method with models trained with different random seeds in Figure 8b. The shaded area represents the min and max values over the three models. We keep the separate curves for different calibration sets' results, since the difference is too small, and it is difficult to see the variation. Note that we have merged old Figures 8 and 9 into the new Figure 8.**
> > >
> > > **We also add a new ablation study in Figure 5b using min, mean, and max values when counting NN ranking across all the examples to better evaluate the effectiveness of using minimum value. This ablation study demonstrates the importance of using the minimum, i.e., which signifies the largest importance a given edge can exhibit over the calibration set.**
> > >
> > > ### It is interesting that SparseGPT performs so poorly compared to magnitude based methods, unlike original paper. Do you have any guess?
> > >
> > > **We thank the reviewer for the question. In the original SparseGPT paper, evaluations were conducted primarily on large language models, where pruning is applied almost exclusively to fully-connected and attention layers, instead of convolution layers.**
> > >
> > > **In contrast, our experiments involve convolutional neural networks, where a substantial portion of parameters lies in convolutional layers. Pruning convolutional layers is significantly more challenging due to spatial coupling and the fact that a parameter multiplies several input dimensions, which makes it difficult to quantify that parameter's importance.}%;  methods tailored to linear-layer reconstruction may not transfer directly without adaptation.**
> > >
> > > **We emphasize that our method is very effective for both fully-connected and convolution layers because it is able to quantify an edge's importance across layers and because it is able to capture the largest importance a given parameter can exhibit over the calibration set (as also demonstrated in the newly included ablation study above). We explained the reason for SparseGPT's performance in Section 6.2 in the revised paper.**

---

> > > > ### Comment · Reviewer_vpzj · 2026-03-06
> > > >
> > > > Thanks that helped a lot!

---

### Review · Reviewer_S2de · 2026-02-16

**Summary Of Contributions:**

This research suggests a new approach to think about how information moves across a neural network. Instead of using typical information-theoretic tools like entropy or mutual information, notions from graph theory and differential geometry, notably Ricci curvature is being used. The fundamental purpose is to find out which connections (weights) in a trained neural network are really critical for how well it works.
'Neural Curvature (NC)' is the key technical innovation. The authors modify curvature in two important ways:
Activation-aware neighbor distributions and Activation-aware edge cost

Strengths:
1. Good conceptual novelty: Using Ricci curvature from differential geometry to analyze neural network information flow is genuinely creative. This paper instead reframes the network as a geometric object and interprets bottlenecks via curvature which is quite novel.
2. Theory backed by mathematical ground: The paper carefully builds Neural graph construction, Data-dependent neighbor distributions, Activation-aware edge cost and Neural curvature definition via α → 1 limit. The definitions are mathematically grounded and extend prior ORC work logically. It’s not just “inspired by curvature” — it rigorously adapts it.
3. Data efficiency: The method requires few calibration samples which in turn removes a potential practical objection.
4. Convincing results from pruning experiments: The pruning experiments convincingly show:
- Removing positive-curvature edges barely affects accuracy.
- Removing negative-curvature edges collapses performance quickly.
The separation is clean and visually strong and this shows the signal is not weak or noisy.

However here are a few weaknesses I found which would be strengthened:
1. Scalability concerns:
- Wasserstein distance requires solving a linear program per edge. Even with optimizations, this is expensive. It scales poorly to large transformers or LLMs. he experiments are limited to small-to-medium CNNs. Can this realistically scale to modern foundation models?
2. Interpretability Claims Are Stronger Than Evidence:
- The paper suggests Symbolic analysis, Model repair, Alignment, Architecture search. But none of these are demonstrated. Right now, curvature identifies important edges — which is useful — but broader claims feel forward-looking rather than proven. This weakens the position a bit.
3. Computational Cost vs Benefit Tradeoff Not Quantified:
We see Curvature sometimes slightly better than magnitude sometimes comparable. But we don’t see Runtime comparison, Compute overhead, Memory footprint
If curvature is much more expensive but only slightly better, that matters.

Overall the paper if of a high intellectual quality, strong novelty, great empirical results but whether it is practically scalable is debatable. The breadth of its scope is also limited.

**Audience:**

Yes

**Audience Explanation:**

People working on:
- Geometric deep learning
- Information flow analysis
- Curvature in graphs

would find this intellectually interesting because it extends curvature ideas to feedforward NNs in a principled way.

**Broader Impact Concerns:**

I do not identify significant direct ethical risks associated with the proposed method. The work introduces a curvature-based framework for analyzing neural network connectivity and importance ranking, which is primarily a methodological and analytical contribution. It does not introduce new data collection mechanisms, sensitive data usage, or deployment-oriented systems.

**Claims And Evidence:**

Yes

**Claims Explanation:**

Yes, the core claims such as the neural curvature can rank connections by importance, and negative-curvature edges are critical for performance, are supported by convincing empirical evidence.
However, some broader positioning claims like negative curvature corresponds to essential data flow are not fully substantiated. It is theoretically only partially supported.
Another claim - This framework enables symbolic analysis, robustness, model repair, alignment is not at all supported by clear evidence.

**Requested Changes:**

1. Length reduction:
The paper is significantly longer than necessary for the contribution. Reduce redundancy in curvature background. Get rid of repeated experimental descriptions.

2. Pruning vs Analysis:
The paper claims it is not primarily a pruning method — yet evaluation is entirely pruning-based. This needs to be adjusted.

3. Add/Improve scalability related points:
The Wasserstein computation is expensive. Defend your analysis on how this will be handled.

4. Add Statistical Robustness:
Currently unclear:
- Are results averaged over multiple seeds?
- What is variance?
Adding:
-Standard deviation across runs
-Error bars in pruning plots
This strengthens empirical rigor.

---

> ### Author Response · Authors · 2026-03-03
> **Response**
>
> ## Length reduction:
> ### The paper is significantly longer than necessary for the contribution. Reduce redundancy in curvature background. Get rid of repeated experimental descriptions.
>
> **Thank you for giving helpful suggestions. The revised version of the paper is significantly shorter (by almost three pages). As noted in the main response, we have adjusted the curvature background and experimental descriptions as per your suggestion. Furthermore, as per other reviewers' suggestions, we have moved various parts, such as the algorithm, proof, some ablation studies and figures, to the appendix as well.**
>
> ## Pruning vs Analysis:
> ### The paper claims it is not primarily a pruning method — yet evaluation is entirely pruning-based. This needs to be adjusted.
>
> **Thank you for pointing this out. In the revised version, we have improved the paper's scope and are now focusing exclusively on post-training pruning using graph curvature. This is reflected both in the paper's new title and updated introduction and problem statement, as noted in the main response.**
>
> ## Add/Improve scalability related points:
> ### Computational Cost vs Benefit Tradeoff Not Quantified: We see Curvature sometimes slightly better than magnitude, sometimes comparable. But we don’t see Runtime comparison, Compute overhead, and Memory footprint. If curvature is much more expensive but only slightly better, that matters.
>
> **Thank you for the helpful suggestions. We have added Table 1 the paper, reporting the runtime and computational cost to provide a direct comparison with other methods. As expected, curvature computation based on the Wasserstein distance is more expensive than other methods, which reflects the additional geometric information it captures.**
>
> **In terms of memory, a fair comparison across methods is challenging as some methods require both GPU and CPU memory, to different extents. The primary memory cost for our method arises from the shortest-path computation, which is currently implemented via dynamic programming in matrix form. This step requires storing intermediate dense matrices and thus dominates the peak memory usage. In practice, this can be alleviated by performing the computation in a block-wise manner via matrix partitioning, so that only submatrices are materialized at a time. This significantly reduces the memory footprint and enables scaling to wider layers.**
>
> **Regarding scalability, there are three potential directions for improving scalability, as noted in the main response and described in a new subsection in the Discussion. First, curvature does not need to be computed for all parameters. A practical strategy is to first apply magnitude-based pruning as a coarse filter to remove clearly unimportant parameters, and then apply the curvature-based criterion only to a smaller subset to obtain a finer-grained importance ranking.
> We applied this strategy on VGG9, CE ReLU as a first step test, and provided the result in Figure 10, and also an updated runtime in Table 1. The hybrid method achieves a significant speedup, and we will explore this at greater depth in future work.**
>
> **Moreover, the current implementation uses the exact Wasserstein distance, which involves solving a linear program per edge and is therefore computationally expensive for very large models. This can be substantially accelerated using the Sinkhorn approximation, which provides an efficient and GPU-friendly estimate of the Wasserstein distance.**
>
> **Finally, the Wasserstein distance computation can be greatly simplified in a number of cases, e.g., in input/output layers where it reduces to a simple inner product.**
>
> ## Add Statistical Robustness: Currently unclear:
> ### Are results averaged over multiple seeds?
>
> ### What is variance? Adding: -Standard deviation across runs -Error bars in pruning plots. This strengthens empirical rigor.
>
> **Thank you for the valuable comments. The calibration set corresponds to the validation split from the training data, and the samples used for curvature computation are randomly drawn from this set and aggregated by label. In the original experiments, we fixed the calibration set to ensure exact reproducibility. All methods are evaluated under the same multi-seed protocol for a fair comparison.**
>
> **To evaluate statistical robustness, we have conducted additional experiments using different random seeds for selecting the calibration samples. We also trained two additional CIFAR-10 models (CE, ReLU) with different initialization seeds. The results show consistent performance across different seeds, indicating that the proposed method is stable with respect to both calibration sampling and model initialization.**

---

### Review · Reviewer_PsCL · 2026-02-17

**Summary Of Contributions:**

This paper analyzes information flow within neural networks using the Ricci curvature concept from graph theory and proposes a method (Neural Curvature) to quantify the importance of edges.

Through pruning experiments, the proposed method demonstrates that connections with negative curvature are central to real-world information flow. Experimental verification demonstrates that the proposed method effectively identifies important connections compared to existing pruning methods.

**Additional Comments:**

N/A

**Audience:**

Yes

**Audience Explanation:**

It is expected to be of interest to researchers interested in neural network interpretability, model analysis, pruning, and structural understanding.

It will be particularly valuable to researchers studying NN from a network structure analysis and theoretical perspective.

**Broader Impact Concerns:**

This study primarily focuses on model structure analysis and understanding efficient models, and its direct negative social impact appears minimal.

However, advances in model compression and efficiency technologies could contribute to the expanded use of large-scale models in the future, making discussions about responsible use crucial in the long term.

**Claims And Evidence:**

Yes

**Claims Explanation:**

Overall, the experimental results are consistent across diverse datasets and model configurations, providing relatively compelling support for the main arguments.

However, the argument would be further strengthened if validation on SotA large-scale models, additional experiments, or analyses were included to address tasks other than vision classification, and computational cost issues.

**Requested Changes:**

We request further discussion or experimental support regarding the proposed method's computational cost and applicability to large-scale models.

The contribution of the proposed method would be more clearly demonstrated if practical applications beyond pruning (e.g., robustness improvement, model lightweighting, etc.) were supplemented.

The generalizability of the method would be further enhanced if sensitivity analyses of certain design choices (e.g., curvature calculation parameter settings) were added.

---

> ### Author Response · Authors · 2026-03-03
> **Response**
>
> 1. We request further discussion or experimental support regarding the proposed method's computational cost and applicability to large-scale models.
>
> **Thank you for the helpful suggestions. We agree that scalability is a major concern for our method. As noted in the main response and described in a new subsection in the Discussion, there are three potential directions for improving scalability. First, curvature does not need to be computed for all parameters. A practical strategy is to first apply magnitude-based pruning as a coarse filter to remove clearly unimportant parameters, and then apply the curvature-based criterion only to a smaller subset to obtain a finer-grained importance ranking.
> We applied this strategy on VGG9, CE ReLU as a first step test, and provided the result in Figure 10, and also an updated runtime in Table 1. The hybrid method achieves a significant speedup, and we will explore this at greater depth in future work.**
>
> **Moreover, the current implementation uses the exact Wasserstein distance, which involves solving a linear program per edge and is therefore computationally expensive for very large models. This can be substantially accelerated using the Sinkhorn approximation, which provides an efficient and GPU-friendly estimate of the Wasserstein distance.**
>
> **Finally, the Wasserstein distance computation can be greatly simplified in a number of cases, e.g., in input/output layers where it reduces to a simple inner product.**
>
> **Moreover, as per the other reviewers' suggestion, we have added a table reporting the runtime and computational cost to provide a direct comparison with magnitude-based pruning. As expected, curvature computation based on the Wasserstein distance is more expensive than other methods, which reflects the additional geometric information it captures. We do believe, however, that the above directions for future work provide a promising avenue for improving our method's scalability.**
>
>
> 2. The contribution of the proposed method would be more clearly demonstrated if practical applications beyond pruning (e.g., robustness improvement, model lightweighting, etc.) were supplemented.
>
> **Thank you for this comment. In the revised version, we have improved the paper's scope and are now focusing exclusively on post-training pruning using graph curvature. This is reflected both in the paper's new title and updated introduction and problem statement, as noted in the main response. Other practical applications, such as robustness improvement, are now discussed in the future work subsection in the Discussion.**
>
> 3. The generalizability of the method would be further enhanced if sensitivity analyses of certain design choices (e.g., curvature calculation parameter settings) were added.
>
> **Thank you for the helpful comments. In the original experiments, we fixed the calibration set to ensure exact reproducibility. All methods were evaluated under the same protocol for a fair comparison.
> In the revised version,  to assess statistical robustness, we conducted additional experiments using different random seeds for selecting the calibration samples. We also trained two additional CIFAR-10 models (CE, ReLU) with different initialization seeds. The results show consistent performance across runs, indicating that the proposed method is stable with respect to both the choice of calibration samples and model initialization. In terms of sensitivity to the proposed concepts of neural neighbor distribution and neural cost, the experiments in Figure 5 demonstrate the utility of both these concepts, as evaluated on the CIFAR-10 CE model.**

---

### Author Response · Authors · 2026-03-03
**Response**

Dear TMLR Action Editor and Reviewers,

We would like to thank the reviewers for their detailed and helpful comments on our paper “Analyzing Neural Network Information Flow Using Differential Geometry". These comments have helped us improve the presentation and overall quality of our work. We have carefully considered all comments and put significant effort into revising our paper accordingly. All additions to the paper are highlighted in red.

The primary issues raised by the reviewers were: 1) evaluation rigor; 2) motivation clarity and paper length; 3) scalability challenges. In the modified version of the paper, we addressed these concerns in the following ways:

1. Evaluation rigor. It was pointed out by multiple reviewers that the experiments do not provide error bars and do not evaluate sensitivity due to changing experimental parameters. Furthermore, reviewers suggested that some comparisons were unfair since we focus on a post-training setting, whereas some of the methods we compare with were designed for an iterative prune-and-retrain setting. To address these concerns, in the revised version, we provide three sets of new experiments:

      - First, we investigate our results' sensitivity to training stochasticity: we train three different cross-entropy models on CIFAR-10 and demonstrate that our method not only performs best among all considered approaches but it also has the most predictable performance;

      - In the second experiment, we sample three different calibration sets used to calculate neural curvatures -- once again, we demonstrate that our method has great robustness to this type of stochasticity;

     - Finally, we provide an additional comparison with SparseGPT [1], a post-training pruning technique based on Hessian matrix analysis. Once again, our method is more effective at identifying the least (and most) important edges as it is able to quantify each edge's importance and thus seamlessly rank edges across layers.

2. Motivation clarity and paper length. All reviewers pointed out that the original claim about identifying data flows was too broad and not supported by the pruning experiments. In the revised paper, we have changed both the title (new title is ``Post-Training Neural Network Pruning using Graph Curvature''), introduction and problem statement to properly scope the paper and focus specifically on post-training pruning using graph curvature. Furthermore, we have reduced the paper length by almost three pages, by implementing the reviewers' suggestions to remove redundancy in the background and experiments, to move large figures, algorithms and proofs to the appendix as well as to clarify confusing definitions.

3. Scalability challenges. We agree that scalability is a major concern for the proposed method's current approach. We will explore three promising directions to mitigate this challenge (as discussed in a new subsection in the Discussion section in the paper), as follows. 1) We will consider a hybrid approach where magnitude-based pruning is used to quickly pre-prune irrelevant weights (as assessed over a validation set) and curvature-based pruning is used for the challenging-to-separate, intermediate-value weights. A proof of concept of this method is provided in the Discussion section where we observe a significant speedup using the hybrid approach, with barely any effect on pruning performance. 2) We will investigate approximations to the Wassertstein distance, which can be computed on the GPU, e.g., through Sinkhorn approximation. 3) We will improve the implementation by noting that the Wasserstein distance can be greatly simplified in a variety of cases, e.g., in input and output layers where it reduces to an inner product.

The revised version has been resubmitted. Individual responses to each reviewer are provided in the comments below. We are looking forward to hearing from you.

Sincerely,
Authors

[1] Elias Frantar and Dan Alistarh. SparseGPT: Massive language models can be accurately pruned in one-shot. In International Conference on Machine Learning, pp. 10323–10337. PMLR, 2023.

---

### Author Response · Authors · 2026-03-05
**New revision**

Dear TMLR Action Editor and Reviewers,

We have made a new revision to the paper in response to the latest comments from Reviewer vpzj. The latest version of the paper has been uploaded.

Sincerely, Authors

---

### Author Response · Authors · 2026-05-13
**Camera-Ready Revision Confirmation**

Dear Editor,

Thank you for your handling of our manuscript and for the positive assessment of the work.

We have completed the requested camera-ready updates, including revisions to further clarify the scope and computational limitations of the proposed method. Specifically, we have:

1. Added the code repository link at the end of the abstract.

2. Clarified in the abstract that this work focuses on small- to medium-sized models.

3. Expanded the discussion in the Introduction (second-to-last paragraph) regarding the computational expense of the proposed curvature-based approach, along with potential mitigation strategies and future directions.

4. Added a new bullet point in the experimental summary of Section 6.1 discussing our exploration of approaches to mitigate the computational overhead.

5. Added an Acknowledgement section.

We believe these revisions improve the clarity and completeness of the final manuscript.

Thank you again for your time and consideration.

Best regards,
Authors

---

### Decision · Action_Editor_Ltig · 2026-05-04

**Recommendation:** Accept with minor revision

**Additional Comments:**

The paper introduces a principled curvature-based post-training pruning criterion and demonstrates its effectiveness over alternative approaches. The revision successfully addressed reviewer concerns by tightening scope, adding robustness experiments, reporting runtime, and including a modern baseline comparison. Two of three reviewers recommended acceptance; the third did not respond despite repeated reminders but had a positive initial review. The main limitation is scalability (the method is expensive and tested only on small-to-medium CNNs and datasets) and I ask the authors to ensure this limitation and the proposed mitigations are discussed prominently.

**Audience:**

Yes

**Audience Explanation:**

The paper connects differential geometry, graph theory, and neural network pruning in a novel way. It may interest researchers in pruning, geometric deep learning, and network analysis.

**Claims And Evidence:**

Yes

**Claims Explanation:**

The core claim that adapted Ollivier-Ricci curvature can rank edges for post-training pruning is well supported across several datasets and model configurations. The revision added robustness experiments, a SparseGPT comparison, and runtime reporting, strengthening the evidence. Both recommending reviewers agreed. Scalability to larger architectures remains undemonstrated but does not undermine the claims actually made.